# CAUSALLY CONSTRAINED DATA SYNTHESIS FOR PRIVATE DATA RELEASE

## ABSTRACT

Data privacy is critical in many decision-making contexts, such as healthcare and finance. A common mechanism is to create differentially private synthetic data using generative models. Such data generation reflects certain statistical properties of the original data, but often has an unacceptable privacy vs. utility trade-off. Since natural data inherently exhibits causal structure, we propose incorporating *causal information* into the training process to favorably navigate the aforementioned trade-off. Under certain assumptions for linear gaussian models and a broader class of models, we theoretically prove that causally informed generative models provide better differential privacy guarantees than their non-causal counterparts. We evaluate our proposal using variational autoencoders, and demonstrate that the trade-off is mitigated through better utility for comparable privacy.

## 1 INTRODUCTION

Automating AI-based solutions and making evidence-based decisions both require data analyses. However, in many situations, the data is sensitive and cannot be published directly. Synthetic data generation, which captures certain statistical properties of the original data, is useful in resolving these issues. However, naive data synthesis may not work: when improperly constructed, the synthetic data can leak information about its sensitive counterpart (from which it was constructed). Several *membership inference* (MI) and *attribute inference* attacks demonstrated for generative models (Mukherjee et al., 2019; Zhang et al., 2020b) eliminate any privacy advantage provided by releasing synthetic data. Therefore, effective privacy-preserving synthetic data generation methods are needed.

The de-facto mechanism used for providing privacy in synthetic data release is that of *differential privacy* (DP) (Dwork et al., 2006) which is known to degrade utility proportional to the amount of privacy provided. This is further exacerbated in tabular data because of the correlations between different records, and among different attributes within a record. In such settings, the amount of noise required to provide meaningful privacy guarantees often destroys utility. Apart from assumptions made on the independence of records and attributes, prior works make numerous assumptions about the nature of usage of synthetic data and downstream tasks to customize DP application (Xiao et al., 2010; Hardt et al., 2010; Cormode et al., 2019; Dwork et al., 2009).

To this end, we propose a mechanism to create synthetic data that is agnostic of the downstream task. Similar to Jordan *et al.* (Jordon et al., 2018), our solution involves training a generative model to provide formal DP guarantees. A key distinction arises as we encode knowledge about the causal structure of the data into the generation process to provide better utility. Our approach leverages the fact that naturally occurring data exhibits causal structure. In particular, to induce favorable privacy vs. utility trade-offs, our main contribution involves encoding the *causal graph* (CG) into the training of the generative model to synthesize data. Considering the case of linear gaussian models, we formally prove that generative models trained with additional knowledge of the causal structure of the specific dataset are more private than their non-causal counterparts. We extend this proof for a more broader class of generative models as well.

To validate the theoretical results on real-world data, we present a novel practical solution utilizing variational autoencoders (VAEs) (Kingma & Welling, 2013). These models combine the advantage of both deep learning and probabilistic modeling, making them scale to large datasets, flexible to fit complex data in a probabilistic manner, and can be used for data generation (Ma et al., 2019; 2020a). Thus, in designing our solution, we train *causally informed* and *differentially private* VAEs. The

CG can be obtained from a domain expert, or learnt directly from observed data (Zheng et al., 2018; Morales-Alvarez et al., 2021) or by using DP CG discovery algorithm (Wang et al., 2020a). The problem of learning the CG itself is important but orthogonal to the goals of this paper.

We evaluate our approach to understand its efficacy both towards improving the utility of downstream tasks, and robustness to an MI attack (Stadler et al., 2020). Further, we aim to understand the effect of *true, partial and incorrect CG* on the privacy vs. utility trade-off. We experimentally evaluate our solution on a synthetic dataset where the true CG is known. We evaluate on real world applications: a medical dataset (Tu et al., 2019), a student response dataset from a real-world online education platform (Wang et al., 2020b), and perform ablation studies using the Lung Cancer dataset (Lauritzen & Spiegelhalter, 1988).

Through our evaluation, we show that models that are causally informed are *more stable* (Kutin & Niyogi, 2012) than associational (either non-causal, or with the incorrect causal structure) models trained using the same dataset. In the absence of DP noise, causal models enhance the baseline utility[1] by $2.42$ percentage points (PPs) on average while non-causal models degrade it by $3.49$ PPs. With respect to privacy evaluation, prior works solely rely on the value of the privacy budget $\varepsilon$. We take this one step further and empirically evaluate resilience to MI. Our experimental results demonstrate the positive impact of causal information in inhibiting the MI adversary's advantage on average. Better still, we demonstrate that DP models that incorporate both complete or even partial causal information are more resilient to MI adversaries than those with purely differential privacy with the exact same $\varepsilon$-DP guarantees.

In summary, the contributions of our work include:

1. A deeper understanding of the advantages of causality through a theoretical result that highlights the privacy amplification induced by being causally informed (§ 3), and insight as to how this can be instantiated (§ 4.1).
2. Empirical results demonstrating that causally constrained (and DP) models are more utilitarian in downstream classification tasks (§ 5.1) and are robust (on average) to MI attacks (§ 5.2).

## 2 PROBLEM STATEMENT & NOTATION

**Problem Statement:** Formally, we define a dataset $D$ to be the set $\{\mathbf{x}_1, \cdots \mathbf{x}_n\}$ of $n$ records $\mathbf{x}_i \in \mathcal{X}$ (the universe of records); each record $\mathbf{x} = (x_1, \cdots, x_k)$ has $k$ attributes (a.k.a variables $X_1, \cdots, X_k$).

We aim to design a procedure which takes as input a *private (or sensitive) dataset* $D_p$ and outputs a *synthetic dataset* $D_s$. The output should have formal privacy guarantees and maintain statistical properties from the input for downstream tasks. Formally speaking, we wish to design $f_\theta : \mathcal{Z} \to \mathcal{X}$, where $\theta$ are the parameters of the method, and $\mathcal{Z}$ is some underlying latent representation for inputs in $\mathcal{X}$. In our work, we wish for $f_\theta$ to provide the guarantee of differential privacy.

**Differential Privacy (Dwork et al., 2006):** Let $\varepsilon \in \mathbb{R}^+$ be the privacy budget, and $H$ be a randomized mechanism that takes a dataset as input. $H$ is said to provide $\varepsilon$-differential privacy (DP) if, for all datasets $D_1$ and $D_2$ that differ on a single record, and all subsets $S$ of the outcomes of running $H$: $\mathbb{P}[H(D_1) \in S] \leq e^\varepsilon \cdot \mathbb{P}[H(D_2) \in S]$, where the probability is over the randomness of $H$.

**Sensitivity:** Let $d \in \mathbb{Z}^+$, $\mathcal{D}$ be a collection of datasets, and define $H : \mathcal{D} \to \mathbb{R}^d$. The $\ell_1$ sensitivity of $H$, denoted $\Delta H$, is defined by $\Delta H = \max\|H(D_1) - H(D_2)\|_1$, where the maximum is over all pairs of datasets $D_1$ and $D_2$ in $\mathcal{D}$ differing in at most one record.

We rely on generative models to enable private data release. If they are trained to provide DP, then any further post-processing (*i.e.,* using them to obtain a *synthetic dataset*) is also DP by post-processing (Dwork et al., 2014). In this work, we use variational autoencoders (VAEs) as our generative models.

**Variational Autoencoders (VAEs)** (Kingma & Welling, 2013): Data generation, $p_\theta(\mathbf{x}|z)$, is realized by a deep neural network (DNN) parameterized by $\theta$, known as the *decoder*. To approximate the posterior of the latent variable $p_\theta(z|\mathbf{x})$, VAEs use another DNN (the *encoder*) with $\mathbf{x}$ as input to produce an approximation of the posterior $q_\phi(z|\mathbf{x})$. VAEs are trained by maximizing an evidence lower bound (ELBO), which is equivalent to minimizing the KL divergence between $q_\phi(z|\mathbf{x})$ and

---

[1]Utility obtained from models trained on the original dataset (without the use of any generative model).

$p_\theta(z|\mathbf{x})$ (Jordan et al., 1999; Zhang et al., 2018). Solutions using VAEs for data generation would concatenate all variables as $X$, train the model, and generate data through sampling from the prior $p(Z)$. To train the model, we wish to minimize the KL divergence between the true posterior $p(z|\mathbf{x})$ and the approximated posterior $q_\phi(z|\mathbf{x})$, by maximizing the ELBO:

$$\text{ELBO} = \mathbb{E}_{q_\phi(z|\mathbf{x}))}[\log p_\theta(\mathbf{x}|z)] - \mathbb{KL}[q_\phi((z|\mathbf{x})||p(z)]$$

**Causally Consistent Models:** Formally, the underlying data generating process (DGP) is characterized by a causal graph that describes the conditional independence relationships between different variables. In this work, we use the term *causally consistent* models to refer to those models that factorize in the causal direction. For example, the graph $X_1 \to X_2$ implies that the factorization following the causal direction is $p(X_1, X_2) = p(X_1) \cdot p(X_2|X_1)$. Due to the *modularity property* (Woodward, 2005), the mechanism to generate $X_2$ from $X_1$ is independent from the marginal distribution of $X_1$. This only holds in causal factorization but not in anti-causal factorization.

## 3    PRIVACY AMPLIFICATION THROUGH CAUSALITY

Here, we present our main result. Stated simply: *under infinite training data, causally consistent (or simply causal) models are more private than their non-causal (or associational) counterparts.* We also characterize the conditions needed for this claim to be true under finite training data. For ease of exposition, we first consider the setting of linear gaussian structural causal models (SCMs) where each node (*i.e.,* variable) is generated as a linear function of its parents in the causal graph (CG).

**Causal and Associational models.** Let $\mathcal{M} = \langle X, f, \epsilon \rangle$ be a linear gaussian SCM corresponding to a CG $G = (X, E_G)$. $X$ is the set of variables $\{x_1, \cdots, x_k\}$, $E_G$ are the edges in the CG connecting them, $f$ represents the linear generating function for each variable $x_i \in X$, and $\epsilon$ are the error terms. We assume all variables are standardized to be zero mean and unit variance. We use upper-case variables to capture sets, bold-faced to capture vectors, and subscripts capture appropriate indexing.

In a linear gaussian SCM, each node is generated as a linear function of its parents (assuming no interaction between them).

$$\boldsymbol{x}_i = \boldsymbol{Pa}_i\boldsymbol{\beta}_i + \boldsymbol{\epsilon}_i \tag{1}$$

where $\boldsymbol{Pa}_i$ is a matrix of parent variables of $x_i$ (of size $n \times k_i$, where $n$ is the number of data points and $k_i \leq k$ denotes the number of parents of $x_i$), and similarly $\boldsymbol{\beta}_i$ is the estimated coefficient vector (of size $k_i \times 1$). The error terms $\epsilon_i$ are mutually independent as well as independent of all other variables. We can also write it as $\boldsymbol{X}\boldsymbol{\beta}_{i,ext} + \boldsymbol{\epsilon}_i$ where $\boldsymbol{X}$ is the $n \times k$ matrix with *all* variables as columns and $\boldsymbol{\beta}_{i,ext}$ is an extended vector such that its value is fixed to 0 for all non-parents of $x_i$.

A **causal generative model** has additional knowledge of the CG. Since mechanisms of the SCM are stable and independent (Peters et al., 2017), fitting the causal generative model can be broken down into a set of separately fit linear regression models. For any variable $x_i$, parameters $\boldsymbol{\beta}_i$ are learnt (as $\hat{\boldsymbol{\beta}}_i$) by minimizing the least squares error, $\ell(\hat{\boldsymbol{x}}_i, \boldsymbol{x}_i) = \sum_{j\in[n]}(\hat{x}_i^j - x_i^j)^2$, where $\hat{\boldsymbol{x}}_i$ is given by,

$$\hat{\boldsymbol{x}}_i = \boldsymbol{Pa}_i\hat{\boldsymbol{\beta}}_i \tag{2}$$

An **associational generative model** does not have knowledge of the true CG. In general, it can be a generative model such as a VAE. However, for learning linear functional relationships, it makes sense to instead learn a set of linear regression equations, which is based on an alternative acyclic structure (*e.g.,* an incorrect graph). For each $x_i$, let $\boldsymbol{H}_i$ be the feature matrix used to predict $x_i$. We obtain,

$$\hat{\boldsymbol{x}}_i = \boldsymbol{H}_i\boldsymbol{\gamma}_i \tag{3}$$

where $\boldsymbol{H}_i$ is the data for all features that generate the value of $x_i$ in the model, analogous to $\boldsymbol{Pa}_i$. For each $x_i$, $\boldsymbol{\gamma}_i$ is the learnt parameter vector of the associational model.

We show that sensitivity of $\boldsymbol{\beta} = \{\boldsymbol{\beta}_1, \cdots, \boldsymbol{\beta}_k\}$ is lower than or equal to $\boldsymbol{\gamma} = \{\boldsymbol{\gamma}_1, \cdots, \boldsymbol{\gamma}_k\}$. To do so, we first prove a result about comparing sensitivity of linear regression when features are chosen by the true data-generating process (DGP) or not. Our result on linear regression can be found in Appendix A (see Lemma 1). Note that since our goal is to compare between models, we follow a different set of assumptions than standard DP on linear regression. Rather than assuming that the inputs are bounded, we assume that the error terms in the DGP are bounded, thus providing a bound on the values of parameters that are optimal for any point.

### 3.1 MAIN THEOREM

**Theorem 1.** *Consider a linear gaussian SCM $\mathcal{M} = \langle X, f, \epsilon \rangle$ with standardized variables (zero mean, unit variance). Let the true generative equations be expressed as,*

$$\forall x_i \in X : \boldsymbol{x}_i = \boldsymbol{Pa}_i \boldsymbol{\beta}_i + \boldsymbol{\epsilon}_i \tag{4}$$

*where (a) $\boldsymbol{Pa}_i$ is the data matrix denoting all parents of $x_i$ in the CG corresponding to $\mathcal{M}$, (b) $\boldsymbol{\beta}_i$ are the true generative parameters, and (c) $\boldsymbol{\epsilon}_i$ is an independent, symmetric error vector that is bounded such that the maximum deviation from $\boldsymbol{\beta}_i$ that minimizes $|\boldsymbol{x}_i^j - \boldsymbol{\beta}_i \boldsymbol{Pa}_i^j|$ for the $j^{th}$ data point is $\delta_{max}$. Let a mechanism take as input $n$ data points and output the parameters of the fitted linear generative functions. Then,*

1. *As $n \to \infty$, the sensitivity of the mechanism based on causal information is lower than that without it.*
2. *For finite $n$, whenever the empirical correlation of a variable's parents with error is much less than the empirical correlation of the features used by the associational model with error (i.e., $\boldsymbol{Pa}_i^T \boldsymbol{\epsilon}_i << \boldsymbol{H}_i^T \boldsymbol{\epsilon}_i^2$), the sensitivity of the mechanism based on causal information is the lowest.*

**Proof idea.**[3] Since a SCM consists of independent mechanisms, it is sufficient to prove the result for regression parameters of any one mechanism. Given a fitted linear regression for one of the nodes in the CG, its sensitivity is determined by stability: *how much a new training point can alter the learnt parameters*; the new point is also constrained to be generated by the linear SCM equation, so the optimal learnt parameters for the new point is the the true SCM parameters (within a $\delta_{max}$ bound).

In the infinite data regime, for a causal model, the fitted parameters are *exactly the true SCM parameters* since it uses exactly the parents as features. Whereas the associational model uses a different set (with possibly correlated features) and therefore the learnt parameters for each feature are different than the SCM parameters. So a new point generated from the true SCM parameters can introduce a significant change in learnt parameters for associational models (note that new input is not bounded), while for causal models the effect is bounded by $\delta_{max}$.

For finite data, we additionally need that the associational model deviates substantially from the causal features (*i.e.,* the contribution of non-causal features to the model is substantial); otherwise a causal model may not have significant improvement on the sensitivity than the associational model.

The above proof also generalizes to case where all variables of SCM are not observed. We need the following assumption.

**Assumption 1.** *For each node $x_i \in X$ for the SCM $\mathcal{M}$, any unobserved parents $Pa_i^{unobs}$ are independent of all other observed parents of $x_i$, $Pa_i^{obs}$.*

$$Pa_i^{j,obs} \perp\!\!\!\perp Pa_i^{k,unobs} \forall j, k \quad \text{where} \; Pa_i^{obs} \cup Pa_i^{unobs} = Pa_i \tag{5}$$

**Corollary 1.** *Under Assumption 1, Theorem 1 is satisfied even if some variables of the SCM are unobserved.*

By assuming strong convexity of the loss function, we are able to generalize our result beyond linear gaussian SCMs; the detailed proof is in Appendix B. The key insight again stems from the higher stability of causal models (in both the infinite and finite data regimes) resulting in lower sensitivity.

**Why does causality provide any privacy benefit?** In the case of causal models, we know the proper factorization of the joint probability distribution. These factorized conditional probabilities remain stable even under new data that corresponds to a different joint distribution (Peters et al., 2017). Hence any learnt model using the factorization will be more *stable*. Due to this higher stability, its (worst-case) loss on unseen points (or points generated by the true DGP) is lower. In the associational case, the model does not have any such constraint, and thus may learn relationships that are not stable.

## 4 OUR APPROACH AND IMPLEMENTATION

Our theoretical analysis shows the privacy benefit of training a causally informed generative model for the linear gaussian case. In practice, the generation of synthetic datasets requires flexibility for

---

[2]$T$ is used to denote the transpose of the matrix.

[3]The detailed proof is in Appendix A.

downstream tasks and computational efficiency in large-scale applications. Therefore, we propose our approach of building causally informed generative models using VAEs and implement this approach for evaluating on real-world datasets; salient features of each dataset is presented in Table 2. Experiments were performed on a server with 8 NVIDIA GeForce RTX GPUs, 48 vCPU cores, and 252 GB of memory. All our code was implemented in `python`.

## 4.1 CAUSAL DEEP GENERATIVE MODELS

Let us consider that the original dataset contains variables $X_1$ and $X_2$ and the causal relationship follows Figure 9(b) (in Appendix C.1). For example, $X_1$ can be a medical treatment, $X_2$ can be a medical test, and $Z$ is the patient health status which is not observed. Instead of using a VAE to generate the data, we design a generative model as shown in Figure 9(c), where the solid line shows the model $p(X_1, X_2, Z) = p(Z) \cdot p_{\theta_1}(X_1|Z) \cdot p_{\theta_2}(X_2|X_1, Z)$, and the dashed line shows the inference network $q_\phi(Z|X_1, X_2)$. In this way, the model is consistent with the underlying CG.

The modeling principle is similar to that of prior work, CAMA (Zhang et al., 2020a). However, CAMA only focuses on prediction and ignores all variables outside the Markov Blanket of the target. In our application, we aim for data generation and need to consider *the full causal graph*, and generalizes CAMA.

**Remark 1:** In this work, we assume that the causal relationship is given. In practice, this can be obtained from domain expert or using a careful chosen causal discovery algorithm (Glymour et al., 2019); the latter can be learnt in a differentially-private manner (Wang et al., 2020a). Additionally, recent advances enable simultaneously learning the CG and optimizing the parameters of a generative model informed by it (Morales-Alvarez et al., 2021). We stress that the contribution of our work is understanding the influence of causal information on the privacy associated with generative models, and not on mechanisms to learn the required causal information.

**Remark 2:** The theory we propose in Appendices A and B show that the privacy amplification induced by causal information is agnostic of the particular choice of generative model. We utilize VAEs as the work of Morales-Alvarez et al. (2021) provides a platform for simultaneously performing (**P1**) causal discovery and (**P2**) causal inference. Recent work provides both the aforementioned properties in the context of generative adversarial networks (GANs) (Geffner et al., 2022) and flow-based models (Kyono et al., 2021), and validating their efficacy is a subject of future research.

## 4.2 PRIVACY BUDGET ($\varepsilon$)

For training our DP models, we utilize `opacus v0.10.0` library[4] that supports the DP-SGD training approach proposed by Abadi et al. (2016) of clipping the gradients and adding noise while training. We ensure that the training parameters for training both causal and non-causal models are fixed. These are described in Appendix C. For all our experiments, we perform a grid search to obtain the optimal clipping norm and noise multiplier. Once training is done, we calculate the privacy budget after training using the Renyi DP accountant provided as part of the `opacus` package. The value of $\varepsilon$ for both causal and non-causal models is the same (refer Table 2).

## 4.3 MEMBERSHIP INFERENCE (MI) ATTACK

Prior solutions for private generative models often use the value of $\varepsilon$ as the sole measure for privacy (Jordon et al., 2018; Zhang et al., 2020b). In addition to $\varepsilon$, we use an MI attack specific to generative models to empirically evaluate if the models we train leak information (Stadler et al., 2020). In this attack, the adversary has access to (a) synthetic data sampled from a generative model trained *with a particular record* in the training data, and (b) synthetic data sampled from a generative model trained *without the same record* in the training data. The objective of the adversary is to use this synthetic data (from both cases) and learn a classifier to determine if a particular record was used during training.

## 4.4 UTILITY METRICS

Utility is preserved if the synthetic data performs comparably to the original sensitive data for *any* given task using *any* classifier. To measure the utility change, we perform the following experiment: if a dataset has $k$ attributes, we utilize $k - 1$ attributes to predict the $k^{th}$ attribute. We randomly choose 20 different attributes to be predicted. Furthermore, we train 5 different classifiers for this task,

---

[4]`https://github.com/pytorch/opacus`

| Dataset (Baseline Utility Range) | DP | Non Causal | | | | | Causal | | | | |
|---|---|---|---|---|---|---|---|---|---|---|---|
| | | kernel | svc | logistic | rf | knn | kernel | svc | logistic | rf | knn |
| EEDI (86-92%) | ✓ | 6.83 | 7.11 | 6.82 | 6.54 | 5.36 | -4.04 | -0.22 | -3.49 | 6.09 | 5.87 |
| | × | 1.14 | 3.86 | 2.6 | 5.27 | 2.41 | -6.83 | -2.74 | -6.13 | -0.03 | -1.96 |
| Pain1000 (88-95%) | ✓ | 9.56 | 6.34 | 2.48 | 5.86 | 2.03 | -1.84 | 4.03 | -1.27 | 2.22 | -4.64 |
| | × | 5.42 | 5.5 | 1.31 | 6.22 | -0.924 | -1.73 | 1.56 | -3.16 | 2.11 | -6.92 |
| Pain5000 (92-98%) | ✓ | 4.09 | 6.89 | 6.8 | 2.27 | 5.22 | 1.7 | 2.11 | 4.58 | 0.62 | -0.82 |
| | × | 3.53 | 5.93 | 5.68 | 0.48 | 4.07 | -0.64 | 0.07 | -0.24 | -4.62 | -5.13 |

Table 1: **Downstream Utility Change:** We report the utility change induced by synthetic data on downstream classification tasks in comparison to the original data *i.e.,* (original data utility - synthetic data utility). Negative values indicate the percentage point improvement, while positive values indicate degradation. The performance range of the classifiers we consider is reported in parentheses next to each dataset. Observe that (a) DP training induces performance degradation in both causal and non-causal settings, and (b) performance degradation in the causal setting is *lower* than that of the non-causal setting.

and compare the predictive capabilities of these classifiers when trained on (a) the original sensitive dataset, and (b) the synthetically generated private dataset. The 5 classifiers are: (a) linear SVC (or kernel), (b) svc, (c) logistic regression, (d) rf (or random forest), (e) knn. Additionally, we draw pairplots using the features from the original and the synthetic dataset to compare their similarity visually. These pairplots are obtained by choosing 10 random attributes (out of the $k$ available attributes). Refer Appendix E for more details.

## 5 Evaluation

Since the accounting mechanism returns the same value of $\varepsilon$, our evaluation is designed to understand:

- If (DP) causally informed models negatively influence accuracy?
- If causally informed models leak more information than their non-causal counterparts?

From our evaluation, we observe that:

- In the absence of DP, causal models enhance the baseline utility computed using the original dataset by $2.42$ percentage points (PPs) on average while non-causal models degrade it by $3.49$ PPs (§ 5.1).
- The influence of causality (with DP) on MI resilience is nuanced. The general trend we observe is that, as the accuracy of causal information increases, so does the model's resilience to MI (§ 5.2).

### 5.1 Utility Evaluation

Table 1 shows the change in utility on different downstream tasks using the generated synthetic data when trained with and without causality as well as DP. The negative values in the table indicate an *improvement* in utility[5]. We only present the range of absolute utility values when trained using the original data in the table and provide individual utility for each classifier in Appendix E.

We observe an average performance degradation of 3.49 percentage points (PPs) across all non-causal models trained without DP, and an average *increase of 2.42* PPs in their causal counterparts. However, it is well understood that DP training induces a privacy vs. utility trade-off, and consequently the utility suffers (compare rows with DP and without DP). However, when causal information is incorporated into the generative model, we observe that the utility degradation is less severe (compare pairs of cells with and without causal information). These results suggest that causal information encoded into the generative process *improves the privacy vs. utility trade-off i.e.,* for the same $\varepsilon$-DP guarantees, the utility for causal models is better than their non-causal counterparts.

In Figure 1, we plot the utility (measured by the average accuracy across the 5 downstream prediction tasks) for both causal and associational models, for varied values of $\varepsilon$. Observe that for a fixed $\varepsilon$, the causal models always have better utility than their associational counterparts. Note that our work is the first to combine DP and causality. When compared with prior work, DP-PGM McKenna et al. (2019) (which uses only DP), we observe that our approach (Causal in Figure 1) generates synthetic data that is more utilitarian than DP-PGM.

---

[5]All trials were reported 5 times with different random seeds. The numbers reported in the table are an average of these trials.

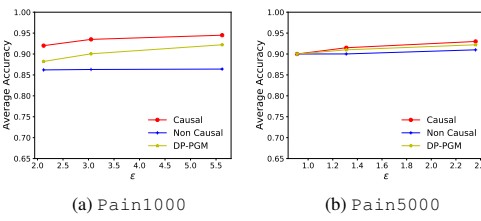

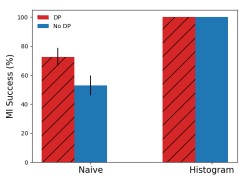 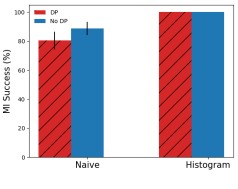

(a) Pain1000      (b) Pain5000

(a) EEDI (Condensed)    (b) EEDI (Morales-Alvarez et al. (2021))

Figure 1: **Utility vs. Privacy:** Causal models always outperform their associational counterparts, for the same $\varepsilon$.

Figure 2: **More Causal Information:** EEDI Morales-Alvarez et al. (2021) is resilient to MI attacks with accurate causal information.

Note that we evaluated the quality of synthetic data generated using a generative model trained using the approach proposed by Morales-Alvarez *et al.* Morales-Alvarez et al. (2021), and the results are not significantly different from those presented in Table 1 (which we omit for brevity).

## 5.2 EVALUATING MI RESILIENCE

Understanding the influence of causal information on MI requires nuanced discussion. Thus far, our discussion has focused on how *perfect* causal information can be used to theoretically minimize $\varepsilon$. However, causal information is often incomplete/partial, or incorrect. Our evaluation answers:

1. What is the effect of complete, partial and incorrect causal information on MI attack accuracy?
2. What is the effect on MI attack accuracy when a causally informed VAE is trained both with and without DP?

We summarize our key results below:

1. Knowledge of a complete CG and training with DP consistently reduces the adversary's advantage across different feature extractors and classifiers *i.e.,* provides better privacy (Figure 3).
2. We observe that incorrect causal information has disparate impacts on resilience to MI. While introducing spurious correlations improves MI efficacy, removing causal information reduces MI efficacy but also degrades utility (§ 5.2.2).
3. Even partial causal information reduces the advantage of the adversary when the model is trained without DP and in most cases, with DP as well. As the accuracy of causal information increases, so does the model's resilience to MI attacks (§ 5.2.3).

**Evaluation Methodology.** We train 2 generative models: one that encodes information from a CG and one that does not. We train each of them with and without DP, and thus have 4 models in total. For all our datasets, we evaluate these models against the MI adversary (§ 4.3), and compute the attack accuracy when a method (DP/causal consistency) is not used in comparison to when it is used. Note that as part of the MI attack, we are unable to utilize the Correlation and Ensemble feature extractors for the EEDI dataset due to computational constraints in our server.

### 5.2.1 WITH COMPLETE CG

We conduct a toy experiment with synthetic data where the complete (true) CG is known apriori. The data is generated based on the CG defined in Appendix D. The results are presented in Figure 3. Here, advantage degradation is the change in MI success between a scenario when DP is not used, and when DP is used for training (positive values are better). Observe that in both the causal and non-causal model, training with DP provides an advantage against the MI adversary, though to varying degrees. However, the important observation is that causal models provide *greater* resilience on average in comparison to the non-causal model.

### 5.2.2 ABLATION STUDY: INCORRECT CG

Many real-world datasets do not come with their associated CGs. CGs obtained from domain experts, or those learnt through algorithmic means are potentially erroneous. We wish to understand how these errors assist MI adversaries. To this end, we utilize the causally informed generative modelling framework of Morales-Alvarez et al. (2021) to synthesize data for 3 different models for the Lung Cancer dataset: (a) armed with the true CG (baseline), (b) with an edge added, and (c) with an edge removed. Observe that the last 2 scenarios are introduced to induce error in the CG. The results of *average MI success* (obtained by averaging the efficacy of the MI attack (Stadler et al., 2020) using 4 classifiers) is presented in Figure 4 (lower is better). By adding edges, we introduce more spurious

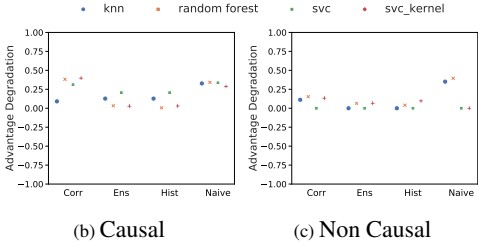

(b) Causal      (c) Non Causal

Figure 3: **True CG:** Both causal and non-causal models trained with DP reduce the adversary's advantage. Causal models provide more advantage degradation on average.

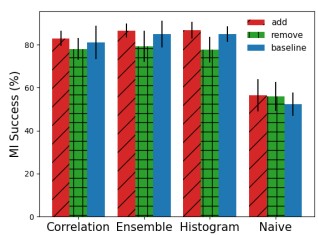

Figure 4: **Incorrect CGs:** For the `Lung Cancer` dataset, adding edges (introducing spurious relationships) enables the MI adversary, but removing edges (disabling causal relationships) hurts it.

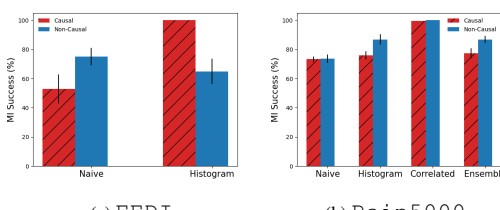

(a) `EEDI`      (b) `Pain5000`

Figure 5: **Partial Causal Information & No DP:** We plot the (average) MI success the adversary uses a non-causal model and switches to its causal counterpart. Observe that even in the absence of DP, causal information by itself provides resilience against our MI adversary.

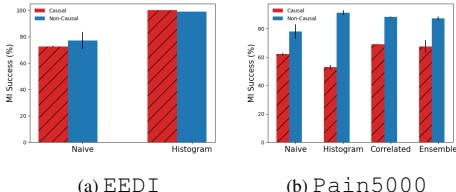

(a) `EEDI`      (b) `Pain5000`

Figure 6: **Partial Causal Information & DP:** We plot the MI success when the adversary uses a non-causal model and switches to its causal counterpart. Observe that in the presence of DP & causal information, the adversary is less effective.

correlations which the MI adversary exploits, but by removing (causal) edges, we remove signal that the MI adversary can use. While removing edges may seem tempting from a privacy perspective, the average (across 8 trials) downstream utility of the data generated from the corresponding model suffers. The utility of the data generated from the true CG is $82.94\%$ which reduces to $81.8\%$ when edge is added and futher to $79.67\%$ when an edge is removed. Our results show that while knowing a true CG is always useful both for privacy and utility, having access to an approximate CG may also provide privacy benefits at the cost of utility.

**Disclaimer:** There are two ways a model may use incorrect causal structure: missing a true parent of a node, or adding an incorrect parent. Corollary 1 already covers the first: with missing parents in the causal model, the result of lower sensitivity for causal models holds as long as Assumption 1 is true. So overall, missing parents is a weaker violation and we may still obtain the same benefits. But if a model adds an incorrect parent, then its sensitivity will definitely increase.

### 5.2.3 WITH PARTIAL CG

For the real world `EEDI` and `Pain` datasets, we follow two approaches: (a) we utilize information from domain experts to partially construct a condensed CG (where several variables are clubbed into a single node of the condensed CG), and (b) we learn a CG from data and simultaneously train a VAE that is informed by it (Morales-Alvarez et al., 2021)–resulting in a larger, yet partial CG. Note that due to space constraints, we only report results for 2 out of the 3 datasets we evaluate. The trends from `Pain1000` are similar to that of `Pain5000` and are in Appendix F.

**Effect of only Causality.** Figure 5 shows the MI adversary's advantage when the model incorporates causal information *in the absence of DP*. Observe that across both datasets, causal models result in more resilient models *i.e.,* lower attack accuracy. While this effect is moderate in the `EEDI` dataset[6] (Figure 5a) (where the `Histogram` features enable a more effective attack in the causal model), this effect is more pronounced in the `Pain5000` dataset (Figure 5b). This suggests that standalone

---

[6]Unlike the other datasets we consider, the EEDI dataset is *sparse*. The `Histogram` attack relies on counting the number of entries for a particular feature to aid in disambiguation, while the `Naive` attack relies on condensing the entire dataset to summary statistics (such as mean, median, mode); we conjecture that sparsity helps the adversary in this case.

causal information provides some privacy guarantees. We conjecture that the MI adversary uses the spurious correlation among the attributes to perform the attack. Even partial causal information is able to eliminate spurious correlation present in the dataset. Hence, a causally informed VAE is also able to reduce the MI adversary's advantage.

**Effect of Causality with DP.** Figure 6 shows the results for models trained using partial causal information with DP. Similar to the earlier results (of causal information and no DP), models trained with DP and causal information are more resilient to MI attacks. These results validate that causality amplifies privacy provided by DP training. We iterate that from results in Figure 1 we show that for the same $\varepsilon$-DP guarantees, models learnt using causal information provide higher utility, thereby providing a promising direction to balance the privacy vs. utility trade-off.

The results thus far rely on domain experts to partially construct a condensed CG. To see if additional causal information provides more resilience, we repeat the experiments with the `EEDI` dataset using the generative model proposed by Morales-Alvarez et al. (2021) that learns a CG from data. We utilize the same training configuration as in the earlier case, resulting in a model learnt with $\varepsilon \approx 13.$, which learns a CG with 57 nodes (compared to the 3 node CG used thus far). From Figure 2, observe that as we provide more accurate causal information to the causal model, its effects on privacy are exacerbated by the presence of DP noise specifically for the attack using `Naive` features. The attack using `Histogram` features in unaffected even with DP providing empirical evidence to recent work that questions the sufficiency of DP training against MI attacks (Humphries et al., 2020).

### 5.3 DISCUSSION

1. **Better understanding of MI.** Yeom et al. (2018) exploit the relationship between MI and overfitting, and propose DP training as a defense through a loose bound. However, recent work suggests that DP training by itself is insufficient (Humphries et al., 2020), and often implemented incorrectly leading to faulty conclusions (Tschantz et al., 2017). Our results also partially corroborate this observation but further exploration is needed to understand the cause of MI in different settings.
2. **Loose privacy budget.** The values of $\varepsilon$ used in our experiment are large. This is a result of the batch size and training duration of the VAEs we use as DP training is expensive. If computation resource is not a constraint then we should be able to get smaller epsilon values. There is extensive discussion by Bhowmick et al. (2018) and Nasr et al. (2021) which discusses different threat models where larger values of $\varepsilon$ are tolerable.
3. **Non-convex loss functions.** VAEs are deep learning models trained to minimize ELBO which is highly non-convex. Research shows that deep learning framework demonstrate for example locally convex properties (Lucas et al., 2021; Littwin & Wolf, 2020) which explains our results.
4. **Overheads:** The work of Morales-Alvarez et al. (2021) discusses an approach to simultaneously learn the CG and train the generative model with said structure. In practice, we observed that causal discovery using this method adds limited overhead ($\sim 100$ epochs of additional training).

## 6 RELATED WORK

**Private Data Generation:** The primary issue associated with private synthetic data generation involves dealing with data scale and dimensionality. Solutions involve using Bayesian networks to add calibrated noise to the latent representations (Zhang et al., 2017; Jälkö et al., 2019), or smarter mechanisms to determine correlations (Zhang et al., 2020b). Utilizing synthetic data generated by GANs has been extensively studied, but only few solutions provide formal guarantees of privacy (Jordon et al., 2018; Wu et al., 2019; Harder et al., 2020; Torkzadehmahani et al., 2019; Ma et al., 2020b; Tantipongpipat et al., 2019; Xin et al., 2020; Long et al., 2019; Liu et al., 2019). Across the spectrum, very limited techniques are evaluated against MI adversaries (Mukherjee et al., 2019).

**Membership Inference:** Most MI work focuses on the discriminative setting (Shokri et al., 2017). More recently, several works propose MI attacks against generative models (Chen et al., 2020; Hilprecht et al., 2019) but offer a limited explanation as to why they are possible. Tople et al. (2020) show the benefits of causal learning to alleviate membership privacy attacks but only limited to classification models and not for generative models.

## 7 CONCLUSIONS

Our work proposes a mechanism for private data release using VAEs trained with differential privacy. Theoretically, we highlight how causal information encoded into the training procedure can potentially *amplify* the privacy guarantee provided by differential privacy, without degrading utility. Empirically, we show how causal information enables advantageous privacy vs. utility trade-offs.

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

# A   DETAILED PROOF: BENEFITS OF CAUSAL LEARNING IN LINEAR GAUSSIAN SCM

## A.1   BACKGROUND

Let $\mathcal{M} = \langle X, f, \epsilon \rangle$ be a linear gaussian structural causal model corresponding to a causal graph $G = (X, E_G)$, where (a) $X$ is the set of variables, $G$ is the causal graph connecting them through edges $E_G$, (b) $f$ represents the set of linear generating functions for each variable $x_i \in X$, and (c) $\epsilon$ are the error terms. We assume that all variables are standardized to be zero mean and unit variance.

In a linear gaussian SCM, each node is generated as a linear function of its parents (we assume without any interaction terms). The error terms $\epsilon$ are mutually independent and independent of all variables.

$$x_i \leftarrow \sum_j \beta_i^j Pa_i^j + \epsilon_i; \quad x_i = Pa_i \beta_i + \epsilon_i \tag{6}$$

where $Pa_i^j$ is a vector referring to the values of $j^{th}$ parent of node $x_i$, $Pa_i$ refers to a matrix of data values with parents of $x_i$ as columns (total $k_i$ columns, where $k_i$ is the number of parents of $x_i$) and $n$ rows as data-points, and $\beta_i$ refers to the true coefficient vector (or structural causal parameter). Alternatively, we can write it as $X\beta_{i,ext} + \epsilon_i$ where $X$ is the matrix with *all* variables as columns and $\beta_{i,ext}$ is an extended vector such that its value is fixed to 0 for all non-parents of $x_i$.

A **causal generative model** has additional knowledge of the graph structure. Since mechanisms of SCM are stable and independent (Peters et al., 2017), fitting the causal generative model can be broken down into a set of separately fit linear regression models. For any variable $x_i$, parameters $\beta_i = \beta_i^1, \beta_i^2 ... \beta_i^{k_i}$ are learnt (as $\hat{\beta}_i$) by minimizing the least squares error, $\ell(x_i, \hat{x}_i) = \sum_{j \in [n]} (\hat{x_i^j} - x_i^j)^2$, where $\hat{x}_i$ is given by

$$\hat{x}_i = \sum_j \hat{\beta}_i^j Pa_i^j; \quad \hat{x}_i = Pa_i \hat{\beta}_i \tag{7}$$

An **associational generative model** does not know the true causal graph, so it may learn an alternative generative acyclic structure, which is also reducible to a set of independently fitted linear regressions. For each $x_i$, let $H_i$ be the matrix denoting the features used to predict $x_i$ (columns are features, rows are data-points). We obtain

$$\hat{x}_i = \sum_j \gamma_i^j H_i^j; \quad \hat{x}_i = H_i \gamma_i \tag{8}$$

where $H_i^j$ and $H_i$ is the individual $j^{th}$ parent data vector and all parents' data matrix respectively, analogous to $Pa_i^j$ and $Pa_i$. $\gamma_i$ is the learnt parameter vector of the associational model, for each $i$

## A.2   GOAL

Our goal is to show that sensitivity of $\hat{\beta} = \{\beta_1, \cdots, \beta_k\}$ is lower than or equal to $\gamma = \{\gamma_1, \cdots, \gamma_k\}$. To do so, we first prove a result about sensitivity of linear regression, which is used to estimate the parameters of the generative model. Note that since our goal is to compare between models, we follow a different set of assumptions than standard differential privacy on linear regression. Rather than assuming that the inputs are bounded, we assume that the error terms in the DGP are bounded, thus providing a bound on the parameter values that are optimal for any point.

Our proof utilizes the following strategy. First, we define 2 worlds: world 1 where where a model is learnt with causal information, and world 2 where a model is learnt without this causal information. Next, we measure the sensitivity of the parameters learnt in both worlds and demonstrate that the sensitivity is lower in world 1 than world 2. This suggests that if DP is used in both worlds, the privacy budget in world 1 will be lower than that of world 2.

## A.3   PRIMER

**Lemma 1.** *Consider dataset $(X^j, y^j)_{j=1}^n$ where the labels are generated by the following equation: $y = \beta X_c + \epsilon$[7] where $X_c \subseteq X$ refers to variables having a non-zero coefficient in the true data-*

---

[7]This equation holds for the data generating distribution; a dataset is sampled from this distribution. More generally, this is the data-generating equation of the SCM that defines the distribution.

*generating process (DGP) of $y$, $\boldsymbol{X}_c$ corresponds to the data matrix associated with variables in $X_c$, and $\boldsymbol{\epsilon}$ is independent error that is bounded and symmetric. $\boldsymbol{\epsilon}$ is bounded such that the maximum deviation from $\boldsymbol{\beta}$ that minimizes $|\boldsymbol{y}^j - \boldsymbol{\beta}\boldsymbol{X}_c^j|$ for any $j^{th}$ data-point is $\delta_{max}$. Consider two linear regression models fit to this dataset, one that has knowledge of the true DGP and includes $X_c$ as features, and one that does not and includes a different subset $X_a \subseteq X$ as features. Assume all variables are standardized (zero mean, unit variance).*

1. *As $n \to \infty$, the sensitivity of linear regression model fitted over $X_c$ is lower than or equal to the sensitivity of the linear regression model fitted over $X_a$.*
2. *For finite $n$, whenever the empirical correlation of $X_c$ with error is much less than other features $X_a$ with error, $X_c\boldsymbol{\epsilon} << X_a\boldsymbol{\epsilon}$, the sensitivity of the mechanism based on causal information is lower than that of the mechanism without it.*

*Proof.* WARM-UP: TWO VARIABLE REGRESSION

As a warm-up exercise, consider a 2-variable regression where $X = \{x_1, x_2\}$. $x_1$ is a part of the true DGP for $y$ (*i.e.,* causes $y$), while $x_2$ is not part of the DGP for $y$ but is related to $y$ (it may be a part of Markov Blanket of $y$, or simply be correlated with $y$). As stated in the Lemma, the corresponding data-generating equation is

$$y = \beta x_1 + \epsilon \tag{9}$$

**(Causal)** $X_c$ **Model.** The model will use the following for predicting $y$,

$$\hat{y} = \hat{\beta}_1 x_1 \tag{10}$$

leading to the following learnt parameter,

$$\hat{\beta}_1 = \frac{\sum x_1 y}{\sum x_1^2} = \frac{\sum x_1(\beta x_1 + \epsilon)}{\sum x_1^2} = \beta + \frac{\sum x_1 \epsilon}{\sum x_1^2} \tag{11}$$

To calculate sensitivity, we assume the existence of an adversary that wishes to add one more point to the training process such that the estimated parameters are farthest from what is currently achieved (capturing the definition of sensitivity).

**Sensitivity.** Let us consider a new data-point added to the training set by an adversary, to maximize difference between $\hat{\beta}_1$ and $\hat{\beta}_1{}'$ (the estimated parameter obtained after adding the adversarial data-point). Any new input chosen by the adversary will be generated based on Eqn 9.

*Note that* the definition of DP is for any 2 databases from a universe of databases; Eqn 9 captures this universe. Since the adversary operates in the same world, any point they sample has to also obey the same equation. Their overall distribution can be different, *e.g.,* $\Pr(y|x_1, x_2)$ can be different, but $\Pr(y|x_1)$ has to remain the same.

Since the error is bounded, the adversary tries to generate a point such that new estimated $\hat{\beta}_1$ is farthest from above. That is, the adversary may choose a point such that the parameter obtained after minimizing the squared loss using that point is $\beta \pm \delta_{max}$ minimizes the squared loss on the point. Further, the adversary can choose a point with large enough $x_1'$ such that estimate on the entire dataset matches $\beta \pm \delta_{max}$. This can be done by choosing a point $(x_1', y')$ such that $|\frac{x_1'\epsilon' + \sum x_1 \epsilon}{x_1'^2 + \sum x_1^2}| = \delta_{max}$.

$\Rightarrow$ Thus, for the parameter corresponding to variable $x_1$, the sensitivity is $|\delta_{max}| + |\frac{\sum x_1 \epsilon_3}{\sum x_1^2}|$. For the parameter corresponding to the variable $x_2$, the sensitivity is zero (since there is no parameter).

**(Associational)** $X_a$ **Model.** In contrast, the full regression model will use the following parameters. Since $x_2$ is not independent of $y$ after conditioning on $x_1$, the model may include $x_2$ since it may result in predictive accuracy gain for $y$.

$$\hat{y} = \gamma_1 x_1 + \gamma_2 x_2 \tag{12}$$

leading to the following learnt parameters,

$$\hat{\gamma_1} = \frac{\sum x_2^2 \sum x_1 y - \sum x_1 x_2 \sum x_2 y}{\sum x_1^2 \sum x_2^2 - (\sum x_1 x_2)^2}$$
$$\hat{\gamma_2} = \frac{\sum x_1^2 \sum x_2 y - \sum x_1 x_2 \sum x_1 y}{\sum x_1^2 \sum x_2^2 - (\sum x_1 x_2)^2} \tag{13}$$

*Sensitivity.* Compared to the $X_c$ model, the second parameter ($\hat{\gamma}_2$) will have non-zero sensitivity (while that for causal model is zero). So we focus on showing that the first parameter $\hat{\gamma}_1$ will also have a higher sensitivity. Observe that the first parameter can be rewritten as

$$\hat{\gamma}_1 = \frac{\sum x_2^2 \sum x_1(\beta x_1 + \epsilon) - \sum x_1 x_2 \sum x_2(\beta x_1 + \epsilon)}{\sum x_1^2 \sum x_2^2 - (\sum x_1 x_2)^2} = \beta + \frac{\sum x_2^2 \sum x_1 \epsilon - \sum x_1 x_2 \sum x_2 \epsilon}{\sum x_1^2 \sum x_2^2 - (\sum x_1 x_2)^2}$$

(14)

Now the adversary can select a $(x_1'', x_2'', y'')$ such that $x_2'' = 0$. Here $x_2$'s coefficient becomes irrelevant and the parameter that minimizes error on any new adversarial point is $\beta \pm \delta_{max}$.

Thus, the sensitivity of the first parameter is

$$|\hat{\gamma}_1 - \beta \pm \delta_{max}| = |\frac{\sum x_2^2 \sum x_1 \epsilon - \sum x_1 x_2 \sum x_2 \epsilon}{\sum x_1^2 \sum x_2^2 - (\sum x_1 x_2)^2}| + |\delta_{max}|$$

Note that this sensitivity can be achieved by choosing $x_1'', y''$ (and therefore $\epsilon''$) for a new adversarial point such that its value is much higher than other points and thus the estimated coefficient tends to $\beta + \delta_{max}$.

We now prove the main claims.

*1. Infinite Data.* As $n \to \infty$, $\sum x_1 \epsilon = 0$ because $x_1$ and $\epsilon$ are independent, by property of the generative process. But $\sum x_2 \epsilon \neq 0$ because it is correlated with $y$. Thus, for infinite data, sensitivity of $X_c$ model ($|\delta_{max}|$) is lower than another $X_a$ model.

*2. Finite Data.* For finite data, if $\sum x_2 \epsilon >> \sum x_1 \epsilon$, then sensitivity of $X_c$ model is lower than or equal to the $X_a$ model.

PROVING THE GENERAL CASE

Using the closed form solution for linear regression, we can write,

$$\hat{\boldsymbol{\beta}} = (\boldsymbol{Z}^T \boldsymbol{Z})^{-1} \boldsymbol{Z}^T \boldsymbol{y}$$

(15)

where $\boldsymbol{Z}$ is a matrix denoting the model's features' values and $\boldsymbol{y}$ is a column vector denoting the values in a dataset for the variable $y$. For the causal model, $\boldsymbol{Z} = \boldsymbol{X}_c$, the true variables from the DGP. Expanding $\boldsymbol{y}$ based on the DGP equation,

$$\hat{\boldsymbol{\beta}} = (\boldsymbol{X}_c^T \boldsymbol{X}_c)^{-1} \boldsymbol{X}_c^T \boldsymbol{y} = (\boldsymbol{X}_c^T \boldsymbol{X}_c)^{-1} (\boldsymbol{X}_c^T \boldsymbol{X}_c)\boldsymbol{\beta} + (\boldsymbol{X}_c^T \boldsymbol{X}_c)^{-1}(\boldsymbol{X}_c^T \boldsymbol{\epsilon})$$
$$\hat{\boldsymbol{\beta}} = \boldsymbol{\beta} + (\boldsymbol{X}_c^T \boldsymbol{X}_c)^{-1}(\boldsymbol{X}_c^T \boldsymbol{\epsilon})$$

(16)

In contrast, for the associational model, we obtain

$$\hat{\boldsymbol{\gamma}} = (\boldsymbol{H}^T \boldsymbol{H})^{-1} \boldsymbol{H}^T \boldsymbol{y}$$
$$= (\boldsymbol{H}^T \boldsymbol{H})^{-1}(\boldsymbol{H}^T \boldsymbol{X}_c)\boldsymbol{\beta} + (\boldsymbol{H}^T \boldsymbol{H})^{-1}(\boldsymbol{H}^T \boldsymbol{\epsilon})$$

(17)

where $\boldsymbol{H}$ represents $\boldsymbol{X}_a$.

The set of features $\boldsymbol{H}$ used by the $X_a$ model may not be equal to the true DGP variables, $X_c$. Thus, the above can be rewritten in terms of the true parameter, $\beta$, as follows,

$$\hat{\boldsymbol{\gamma}} = (\boldsymbol{H}^T \boldsymbol{H})^{-1} \boldsymbol{H}^T (\boldsymbol{X}_c \boldsymbol{\beta}) + (\boldsymbol{H}^T \boldsymbol{H})^{-1}(\boldsymbol{H}^T \boldsymbol{\epsilon})$$
$$= (\boldsymbol{H}^T \boldsymbol{H})^{-1} \boldsymbol{H}^T (\boldsymbol{H}\boldsymbol{\beta}_{i,extended} + \boldsymbol{S}\boldsymbol{\beta}') + (\boldsymbol{H}^T \boldsymbol{H})^{-1}(\boldsymbol{H}^T \boldsymbol{\epsilon})$$
$$= \boldsymbol{\beta}_{i,extended} + (\boldsymbol{H}^T \boldsymbol{H})^{-1} \boldsymbol{H}^T \boldsymbol{S}\boldsymbol{\beta}' + (\boldsymbol{H}^T \boldsymbol{H})^{-1}(\boldsymbol{H}^T \boldsymbol{\epsilon})$$

(18)

where $\boldsymbol{\beta}_{i,extended}$ and $\boldsymbol{\beta}'$ are simply re-parameterizations of the true $\boldsymbol{\beta}$; they are zero for all variables $x \notin X_c$. $\boldsymbol{\beta}_{i,extended}$ is an extension of $\boldsymbol{\beta}$ for variables in $\boldsymbol{H}$, and $\boldsymbol{\beta}'$ is an additional vector which is used only if $\boldsymbol{H}$ does not include all true DGP variables $X_c$. $\boldsymbol{S}$ is the matrix denoting data for all $x \in X_c$ that are not in $\boldsymbol{H}$

PROOF OF MAIN CLAIM

**Infinite data case.** Error is independent of the DGP variables, *i.e.*, $\boldsymbol{X}_c^T \boldsymbol{\epsilon} = 0$ as the number of samples $n \to \infty$. Thus, $\hat{\boldsymbol{\beta}} = \boldsymbol{\beta}$ and any new training point provided by the adversary will also be generated using the same true $\boldsymbol{\beta}$ (within an bound of $\pm \delta_{max}$). Thus, $\hat{\boldsymbol{\beta}} = \boldsymbol{\beta} \pm \delta_{max}$ will also be optimal for this new point and the estimated value will change within $\delta_{max}$.

For the $X_a$ model, however, note that $\hat{\boldsymbol{\gamma}} \neq \boldsymbol{\beta}$ unless $\boldsymbol{H} = \boldsymbol{X}_c$. We provide a construction for adversary's input such that estimated $\hat{\boldsymbol{\gamma}}$ will change more than or equal to $\delta_{max}$ after adding that input. For all variables $x_k \in X_a$ that are correlated (not in $X_c$), an adversary chooses value of $x_k = 0$ and generates $y$ using $\boldsymbol{\beta}$ such that the correlation between $y$ and $x_k$ is broken. Further, for the $X_c$ features and $y$, the adversary is constrained to choose an input such that $\boldsymbol{\gamma}_{X_c} = \boldsymbol{\beta} \pm \delta_{max}$ is optimal on the input, irrespective of the value of other $\boldsymbol{\gamma}$ dimensions (since $x_k = 0$). That is, after addition of new point, the least squares optimization will ignore $x_k$ and move $\hat{\boldsymbol{\gamma}}_{X_c}$ closer to $\boldsymbol{\beta}$ for the $X_c$ features. The total sensitivity is $|\hat{\boldsymbol{\beta}}_{X_c} - \boldsymbol{\beta}| + |\delta_{max}|$ for the parameters corresponding to $X_c$. For all other parameters, the $X_c$ model outputs value of zero (and hence sensitivity of zero), which is trivially lower than or equal to the $X_a$'s model's parameter sensitivity for those parameters.

**Finite data case.** In the finite data case, given a fitted $\hat{\boldsymbol{\beta}}$ the sensitivity that a new adversarial point can lead to is $|\boldsymbol{\beta} \pm \delta_{max} - \hat{\boldsymbol{\beta}}| = |(\boldsymbol{X}_c^T \boldsymbol{X}_c)^{-1} (\boldsymbol{X}_c^T \boldsymbol{\epsilon})| + |\delta_{max}|$ (from Eqn 16). For large-enough $n$, $\boldsymbol{X}_c^T \boldsymbol{\epsilon}$ should be close to zero due to independence. Sensitivity of coefficients for all non-parents is zero. For the $X_a$ model, the coefficients can be written as,

$$\hat{\boldsymbol{\gamma}} = \boldsymbol{\beta}_{i,extended} + (\boldsymbol{H}^T \boldsymbol{H})^{-1} \boldsymbol{H}^T \boldsymbol{S} \boldsymbol{\beta}' + (\boldsymbol{H}^T \boldsymbol{H})^{-1} (\boldsymbol{H}^T \boldsymbol{\epsilon}) \tag{19}$$

As we can see, for the variables $\notin X_c$, $\hat{\boldsymbol{\gamma}}$ depends on the data and therefore will have a non-zero sensitivity to a new data-point, greater than the $X_c$ model.

We next consider sensitivity of parameters corresponding to variables in $X_c$. As with the infinite data case, to generate a new input, the adversary can set the value of variables such that the non-$X_c$ features in $h \in H$ become 0. Then optimal $\hat{\boldsymbol{\gamma}}_{X_c}$ for the new point will be $\beta \pm \delta_{max}$ (which is equivalent to $\boldsymbol{\beta}_{extended} \pm \delta_{max}$), and sensitivity will be the last two terms from Eqn 19 plus $\delta_{max}$ (which can be achieved by sufficiently high values of $X_c$ variables for the adversarial point). Thus, sensitivity is $|\boldsymbol{\beta} \pm \delta_{max} - \hat{\boldsymbol{\gamma}}_{X_c}| = |(\boldsymbol{H}^T \boldsymbol{H})^{-1} \boldsymbol{H}^T \boldsymbol{S} \boldsymbol{\beta}' + (\boldsymbol{H}^T \boldsymbol{H})^{-1} (\boldsymbol{H}^T \boldsymbol{\epsilon})| + |\delta_{max}|$. Since $\boldsymbol{\beta}'$ and $\boldsymbol{S}$ correspond to the true $\boldsymbol{\beta}$ and $\boldsymbol{X}_c$ respectively, , the sensitivity of $X_a$ model will be higher than $X_c$ model's sensitivity for the same parameter whenever $\boldsymbol{H}^T \boldsymbol{\epsilon} >> \boldsymbol{X}_c^T \boldsymbol{\epsilon}$. $\qquad \square$

We now use this lemma to prove our main theorem.

A.4 MAIN THEOREM

**Theorem 1.** *Consider a linear gaussian SCM $\mathcal{M} = \langle X, f, \epsilon \rangle$ with standardized variables (zero mean, unit variance). Let the true generative equations be expressed as,*

$$\forall x_i \in X : \boldsymbol{x}_i = \boldsymbol{Pa}_i \boldsymbol{\beta}_i + \boldsymbol{\epsilon}_i \tag{4}$$

*where (a) $\boldsymbol{Pa}_i$ is the data matrix denoting all parents of $x_i$ in the CG corresponding to $\mathcal{M}$, (b) $\boldsymbol{\beta}_i$ are the true generative parameters, and (c) $\boldsymbol{\epsilon}_i$ is an independent, symmetric error vector that is bounded such that the maximum deviation from $\boldsymbol{\beta}_i$ that minimizes $|\boldsymbol{x}_i^j - \boldsymbol{\beta}_i \boldsymbol{Pa}_i^j|$ for the $j^{th}$ data point is $\delta_{max}$. Let a mechanism take as input $n$ data points and output the parameters of the fitted linear generative functions. Then,*

1. *As $n \to \infty$, the sensitivity of the mechanism based on causal information is lower than that without it.*
2. *For finite $n$, whenever the empirical correlation of a variable's parents with error is much less than the empirical correlation of the features used by the associational model with error (i.e., $\boldsymbol{Pa}_i^T \boldsymbol{\epsilon}_i << \boldsymbol{H}_i^T \boldsymbol{\epsilon}_i$[8]), the sensitivity of the mechanism based on causal information is the lowest.*

*Proof.* **WARM-UP: Three node SCM.** As a warm-up, consider a 3-node SCM where $X = \{x_1, x_2, x_3\}$. $x_1$ causes $x_3$, while $x_2$ does not have a causal relationship with $x_3$ but is related

---

[8]$T$ is used to denote the transpose of the matrix.

to $x_3$ (it may be a part of Markov Blanket of $x_3$, or simply be correlated with $x_3$. The corresponding data-generating equations are,

$$x_1 = \epsilon_1$$
$$x_2 = f_2(x_1, x_3, \epsilon_2) \tag{20}$$
$$x_3 = \beta_3 x_1 + \epsilon_3$$

where $f_2$ can be any linear function ($x_2$ optionally is caused by $x_1$). If $x_2$ depends on $x_3$, it will be a child of $x_3$; if $x_2$ depends on $x_1$, it will be correlated with $x_3$; if $x_2$ depends on both $x_1$ and $x_3$, it will be both a child and correlated with $x_3$. Let us consider the estimation equation for any $x_i$. The proof logic follows the same way for other estimating equations, since they correspond to independent mechanisms.

**Causal Model.** The causal model will use the following model for predicting $x_3$,

$$\hat{x_3} = \hat{\beta}_3 x_1 \tag{21}$$

**Associational Model.** In contrast, the associational generative model will use the following parameters. Since $x_2$ is not independent of $x_3$ after conditioning on $x_1$, the model may include $x_2$ since it may result in predictive accuracy gain for $x_3$.

$$\hat{x_3} = \gamma_3^1 x_1 + \gamma_3^2 x_2 \tag{22}$$

Using Lemma 1, we see that causal model corresponds to the true DGP, whereas associational model does not. Thus, sensitivity of causal model is lower than or equal to associational model, under the same conditions (where $X_c$ is replaced by causal parents $Pa = \{x_1\}$ and $X_a$ by $H = \{x_1, x_2\}$).

We can analogously use Lemma 1 for the general case with all variables. □

The above proof also generalizes to case where all variables of SCM are not observed. We need the following assumption.

**Assumption 1.** *For each node $x_i \in X$ for the SCM $\mathcal{M}$, any unobserved parents $Pa_i^{unobs}$ are independent of all other observed parents of $x_i$, $Pa_i^{obs}$.*

$$Pa_i^{j,obs} \perp\!\!\!\perp Pa_i^{k,unobs} \forall j, k \quad where \ \ Pa_i^{obs} \cup Pa_i^{unobs} = Pa_i \tag{5}$$

If all parents of a node $x_i$ are observed, the above assumption is true trivially. If all parents are not observed (*e.g.*, $x_i$ shares an unobserved common cause with another variable), then this assumption ensures that the estimated error from linear regression is independent of the true parents.

**Corollary 2.** *Under Assumption 1, Theorem 1 is satisfied even if some variables of the SCM are unobserved.*

*Proof.* Under Assumption 1, the empirical error is still independent of the causal parents. Under the simple case, if $x_3$ has an unobserved parent, it may share an unobserved parent with $x_2$ and thus the $\epsilon_3$ term can be expanded as, $\epsilon_3 = \beta_3^{unobs} x_{unobs} + \epsilon_3'$ where $\epsilon_3'$ is the true SCM error with an unobserved variable. However, due to Assumption 1, $x_{unobs} \perp\!\!\!\perp x_1$; hence $x_1 \perp\!\!\!\perp \epsilon_3$. Hence the logic of above proof holds and the rest of the proof follows identically to Theorem 1. □

**REMARK: Counter-Examples Where Causal Information May Not Help Sensitivity.** Note that the above proof would not work if the additional features used by the associational model are independent or weakly correlated with the variable to be generated (and thus we cannot claim that $\boldsymbol{H}_i^T \boldsymbol{\epsilon}_i >> \boldsymbol{Pa}_i^T \boldsymbol{\epsilon}_i$). In the 3-node SCM, for example, if $x_2$ is weakly correlated with $x_3$; $\sum x_2 \epsilon_3$ will also be close to zero, as would be $\sum x_1 \epsilon_3$, for large $n$. So the sensitivity of $\hat{\gamma}_3^1$ may be comparable to $\hat{\beta}_3^1$ (though parameter $\hat{\gamma}_3^2$ may still have non-zero sensitivity compared to the causal model's zero sensitivity). Specifically, if relationship between $x_2$ and $x_3$ is weak enough such that $\sum x_2 \epsilon_3$ is comparable to $\sum x_1 \epsilon_3$, then it is not guaranteed that causal information will help.

To provide a contrived example where a model with causal information will have worse sensitivity than without, consider the following *counter-example*. Suppose a finite dataset such that $\sum x_2^2 \sum x_1 \epsilon_3 -$

$\sum x_1 x_2 \sum x_2 \epsilon_3 = 0$. Then $\hat{\gamma_3}^1 = \beta_3$ and will have minimal ($\delta_{max}$) sensitivity, while $\hat{\beta_3}^1 \neq \beta_3$ and thus its sensitivity is $> \delta_{max}$. In addition, we can construct relationship of $x_2$ and $x_3$ such that sensitivity of $\gamma_3^2$ is also zero. Specifically, $x_2$ may be the child of $x_3$ and be fully determined by $x_3$; $x_2 = \beta_2 x_3 \Rightarrow x_3 = \frac{1}{\beta_2} x_2$, leading to $\hat{\gamma_3}^2 = \frac{1}{\beta_2}$. Then $\hat{\gamma_3}^2$ will also be optimal for any new adversarial point. Thus, comparing the causal versus associational model, the sensitivity of both $\hat{\gamma_3}^2$ and $\hat{\beta_3}^2$ is zero, but the sensitivity of $\hat{\gamma_3}^1$ is lower than that of $\hat{\beta_3}^1$.

**Correctness of the Proof:** Our proof does not simply depend on the number of features. As a counterexample, consider a graph where a node $x_3$ has two parents: $x_1$ and $x_2$ and is correlated with $x_4$. A causal model will use both $x_1$ and $x_2$ while the associational model may use $x_4$. The logic in the proof (independence of error w.r.t features) will show that causal model with more features is less sensitive.

**Clarification (& Comparison to Prior Work)**: We would like to clarify that the proof involving linear mixture of gaussians is meant to serve as intuition to understand what the benefit of causal side-information has towards the stability of the learnt parameters. Prior work (Chaudhuri et al., 2011) also studies the learning of linear models using DP and performs a similar style of analysis. However, we remark that (a) their work does not focus on using any causal information, and (b) measuring sensitivity through parameter stability is a fairly general technique used (and does not imply that output perturbation is the only mechanism to be used to achieve DP in such a setting). We would also like to stress that (a) we use a different proof technique that is motivated by the causal graph structure, and (b) their work is for discriminative models while ours is for generative models.

## B  PRIVACY BENEFIT OF CAUSAL MODELS IN NON-LINEAR SETTINGS

**Note:** The notation in this section is slightly different from that used earlier.

### B.1  NOTATION

A mechanism $H$ takes in as input a dataset $D$ and outputs a parameterized model $f_\theta$, where $\theta$ are the parameters of the model. The model (and its parameters) belongs to a hypothesis space $\mathcal{H}$. The dataset comprises of samples, where each sample $\mathbf{x} = (x_1, \cdots, x_k)$ comprise of $k$ features. To learn the model, we utilize the empirical risk mechanism (ERM), and a loss function $\mathcal{L}$. The subscript of the loss function denotes what the loss is calculated over. For example $\mathcal{L}_\mathbf{x}$ denotes the loss being calculated over sample $\mathbf{x}$. Similarly, $\mathcal{L}_D$ denotes the average loss calculated over all samples in the dataset *i.e.*, $\mathcal{L}_D = \frac{1}{|D|} \sum_{\mathbf{x} \in D} \mathcal{L}_\mathbf{x}$. Additionally, $\mathcal{L}_\mathbf{x}(f_\theta) = \ell(f_\theta(\mathbf{x}), f^*(\mathbf{x}))$ where $f^*$ is the oracle (responsible for generating ground truth), and $\ell(.,.)$ can be any loss function (such as the cross entropy loss or reconstruction loss for a generative model).

**1. Data Generating Process (DGP):** The DGP $\langle f^*, \eta \rangle$ is obtained using the following procedure: $f^* = \lim_{n \to \infty} \arg\min \mathcal{L}_D(f_\theta)$. Essentially $f^\star$ can be thought of as the *infinite data* limit of the ERM learner and can be viewed as the *ground truth*. In a causal setting, the DGP for all variables/features $\mathbf{x}$ is defined as $f^*(\mathbf{x}) = (f_1^*(Pa(x_1)) + \eta_i, \cdots, f_n^*(Pa(x_n)) + \eta_n)$ where $\eta_i$ are mutually, independently chosen noise values and $Pa(x_i)$ are the parents of $x_i$ in the SCM.

**2. Distinction between Causal and Associational Worlds:** For each feature $x_i$, we call $Pa(x_i)$ as the *causal* features, and $X \setminus \{x_i, Pa(x_i)\}$ as the *associational* features for predicting $x_i$. Correspondingly, the model using only $Pa(x_i)$ for each feature $x_i$ is known as the causal model, and the model using all features $X$ (including associational features) is known as the associational model. We denote the causal model learnt by ERM with loss $\mathcal{L}$ as $f_{\theta_c}$, and the associational model learnt by ERM using the same loss $\mathcal{L}$ as $f_{\theta_a}$. Note that the hypothesis class for the models is different: $f_{\theta_c} \in \mathcal{H}_C$ and $f_{\theta_a} \in \mathcal{H}_A$, where $\mathcal{H}_C \subseteq \mathcal{H}_A$.

Like $f_{\theta_c}$, the true DGP function uses only the causal features. Assuming that the true function $f^*$ belongs in the hypothesis class $\mathcal{H}_C$, we write, $f^* = \lim_{|D| \to \infty} \arg\min_{f \in \mathcal{H}_C} \mathcal{L}_D(f)$.

**3. Adversary**. Given a dataset $D$ and a model $f_\theta$, the role of an adversary is to create a neighboring dataset $D'$ by adding a new point $\mathbf{x}'$. We assume that the adversary does so by choosing a point $\mathbf{x}'$ where the loss of $f_\theta$ is maximized. Thus, the difference of the empirical loss on $D'$ compared to $D$ will be high, which we expect to lead to high susceptibility to membership inference attacks.

**4. Loss-maximizing (LM) Adversary**: Given a model $f_\theta$, dataset $D$, and a loss function $\mathcal{L}$, an LM adversary chooses a point $\mathbf{x}'$ (to be added to $D$ to obtain $D'$) as $\arg\max_{\mathbf{x}} \mathcal{L}_{\mathbf{x}}(f_\theta)$. Note that $\mathcal{L}_{\mathbf{x}}(f_\theta) = \mathcal{L}_{\mathbf{x}}(f_\theta(\mathbf{x}))$

---

**Main Result.** Given a dataset $D$ of size $n$, and a strongly convex and Lipschitz continuous loss function $\mathcal{L}$, assume we train two models in a differentially private manner: a causal (generative) model $f_{\theta_c}$, and an associational (generative) model $f_{\theta_a}$, such that they minimize $\mathcal{L}$ on $D$. Assume that the class of hypotheses $\mathcal{H}$ is expressive enough such that the true causal function lies in $\mathcal{H}$.
1. *Infinite sample case.* As $n \to \infty$, the privacy budget of the causal model is lower than that of the associational model *i.e.,* $\varepsilon_c \leq \varepsilon_a$.
2. *Finite sample case.* For finite $n$, assuming certain conditions on the associational models learnt and $n$, the privacy budget of the causal model is lower than that of the associational model *i.e.,* $\varepsilon_c \leq \varepsilon_a$.

---

### B.2 PROOF OUTLINE

The main steps of our proof are as follows:

1. We instantiate two worlds: one with a causal model, and one without one (*i.e.,* with an associational model).
2. We show that the maximum loss of a causal model is lower than or equal to maximum loss of the corresponding associational model.
3. Using strong convexity and Lipschitz continuity of the loss function, we show how the difference in loss corresponds to the sensitivity of the learning function.
4. Finally, the privacy budget $\varepsilon$ is a monotonic function of the sensitivity.

We prove step 2 separately for $n \to \infty$ (Appendix B.1) and finite $n$ (Appendix B.2) below. Then we prove step 3 in Appendix B.3. Step 4 follows from differential privacy literature (Dwork et al., 2014).

### B.3 PROOF WHEN $n \to \infty$ (FOR STEP 2 FROM OUTLINE)

As $|D| = n \to \infty$, the proof arises from knowledge that the causal model becomes the same as the true DGP $f^*$.

**Preliminary 1.** Given any variable $x_t$, the causal model learns a function based only on its parents, $Pa(x_t)$. The adversary for causal model chooses points from the DGP $\langle f^*, \eta \rangle^9$ s.t.,

$$\mathbf{x}' = \arg\max_{\mathbf{x}} \mathcal{L}_{\mathbf{x}}(f_{\theta_c}(\mathbf{x})) \text{ s.t. } \forall i \ \ x_i = f_i^*(Pa(x_i)) + \eta_i \tag{23}$$

where $f_{\theta_c} = \arg\min_{f \in \mathcal{H}_C} \mathcal{L}_D(f)$. Assuming that $\mathcal{H}_C$ is expressive enough such that $f^* \in \mathcal{H}_C$, as $n = |D| \to \infty$, we can write

$$\lim_{|D| \to \infty} f_{\theta_c} = \lim_{|D| \to \infty} \arg\max_{f \in \mathcal{H}_C} \mathcal{L}_D(f) = f^*$$

Therefore, the causal model is equivalent to the true DGP's function. For any target $x_i$ to be predicted, maximum error on any point is $\eta_i$ for the $\ell_1$ loss, and a function of $\eta_i$ for other losses. Intuitively, the adversary is constrained to choose points at a maximum $\eta_i$ distance away from the causal model.

But for associational models, we have,

$$\mathbf{x}'' = \arg\max_{\mathbf{x}} \mathcal{L}_{\mathbf{x}}(f_{\theta_a}(\mathbf{x})) \text{ s.t. } \forall i \ \ x_i = f_i^*(Pa(x_i)) + \eta_i \tag{24}$$

As $n = |D| \to \infty$, $f_{\theta_a} \neq f^*$. Thus, the adversary is less constrained and can generate points for a target $x_i$ that are generated from a different function than the associational model. For any point, the difference in the associational model's prediction and the true value is $|f_{\theta_a}(\mathbf{x})) - f^*(Pa(x_i))| + \eta_i$, which is equivalent to the loss under $\ell_1$. For a general loss function, the loss is a function of $|f_{\theta_a}(\mathbf{x}) - f^*(Pa(x_i))| + \eta_i$. Therefore, we obtain,

$$\forall i \ \ \eta_i \leq |f_{\theta_a}(\mathbf{x})) - f^*(Pa(x_i))| + \eta_i \Rightarrow \max_{\mathbf{x}} \mathcal{L}_{\mathbf{x}}(f_{\theta_c}(\mathbf{x})) \leq \max_{\mathbf{x}} \mathcal{L}_{\mathbf{x}}(f_{\theta_a}(\mathbf{x})) \tag{25}$$

for all losses that are increasing functions of the difference between the predicted and actual value.

---

[9]One should think of the DGP = $\langle f^*, \eta \rangle$ as the oracle that generates labels.

### B.4 PROOF WHEN FINITE $n$ (FOR STEP 2 FROM OUTLINE)

When $n$ is finite, the proof argument remains the same but we need an additional assumption on the associational model $f_{\theta_a}$ learnt from $D$. From learning theory (Shalev-Shwartz & Ben-David, 2014), we know that the loss of $f_{\theta_c}$ will converge to that of $f^*$, while loss of $f_{\theta_a}$ will converge to loss of $f_{\theta_a}^\infty \neq f^*$. Thus, with high probability, $f_{\theta_c}$ will have a lower loss w.r.t. $f^*$ than $f_{\theta_a}$ and a similar argument follows as for the infinite-data case. However, since this convergence is probabilistic and depends on the size of $n$, it is possible to obtain a $f_{\theta_c}$ that has a higher loss w.r.t. $f^*$ compared to $f_{\theta_a}$.

Therefore, rather than assuming convergence of $f_{\theta_c}$ to $f^*$, we instead rely on the property that the true DGP function $f^*$ does not depend on the associational features $x_a$. As a result, even if the loss of the associational model is lower than the causal model on a particular point $\mathbf{x} = x_c \cup x_a$[10], we can change the value of $x_a$ to obtain a higher loss for the associational model (without changing the loss of the causal model). This requires that the associational model have a non-trivial contribution from the associational (non-causal) features, sufficient to change the loss. We state the following assumption.

**Assumption 2:** If $f_{\theta_c}$ is the causal model and $f_{\theta_a}$ is the associational model, then we assume that the associational model has non-trivial contribution from the associational features. Specifically, denote $x_c$ as the causal features and $x_a$ as the associational features, such that $\mathbf{x} = x_c \cup x_a$. We define any two new points: $\mathbf{x}' = x_c' \cup x_a$ and $\mathbf{x}'' = x_c'' \cup x_a''$. Let us first assume a fixed value of $x_a$. The LHS (below) denotes the max difference in loss between $f_{\theta_c}$ and $f_{\theta_a}$ (*i.e.*, change in loss between causal and associational models over the same causal features). The RHS (below) denotes difference in loss of $f_{\theta_a}$ between $x_a$ and another value $x_a^*$, keeping $x_c$ constant (*i.e.*, effect due to the associational features). Formally speaking

$$
\begin{aligned}
\exists x_a \quad & \max_{x_c'}\{\mathcal{L}_{\mathbf{x}'}(f_{\theta_c}(x_c' \cup x_a)) - \mathcal{L}_{\mathbf{x}'}(f_{\theta_a}(x_c' \cup x_a))\} \\
& \leq \min_{x_c''} \max_{x_a''} \mathcal{L}_{\mathbf{x}''}(f_{\theta_a}(x_c'' \cup x_a'')) - \mathcal{L}_{x_c'' \cup x_a}(f_{\theta_a}(x_c'' \cup x_a))
\end{aligned}
\tag{26}
$$

The inequality above can be interpreted as follows: if adversary 1 aims to find the $x_c'$ such that difference in loss between associational and causal features is highest for a given $x_a$, then there can always be another adversary 2 who can obtain a bigger difference in loss by changing the associational features (from the same $x_a$ to $x_a''$).

*Intuition*: Imagine that $f_{\theta_c}$ is trained initially, and then associational features are introduced to train $f_{\theta_a}$. $f_{\theta_a}$ can obtain a lower loss than $f_{\theta_c}$ by using the associational features $x_a$. In doing so, it might even change the model parameters related to $x_c$. Assumption 1 says that change in $x_c$'s parameters is small compared to the importance of the $x_a$'s parameters in $f_{\theta_a}$. For example, consider a $f^*, f_{\theta_c}, f_{\theta_a}$ to predict the value of $x_t$ such that $x_c = \{x_1\}$ and $x_a = \{x_2\}$, and consider $\ell_1$ loss.

$$
f^* = x_1; \quad f_{\theta_c} = 2x_1; \quad f_{\theta_a} = 1.9x_1 + \phi(x_2)
$$

where $x_t = f^*(\mathbf{x}) + \eta$ and $\eta \in [-0.5, 0.5]$. Note that without $\phi(x_2)$, the loss of the associational model is lower than the loss of causal model on any point. However, if $x_a = x_2 \in \mathbb{R}$, then we can always set $|x_2|$ to an extreme value such that $\phi(x_2)$ overturns the reduction in loss for the associational model, without invoking Assumption 1. When $x_a$ is bounded (*e.g.*, $x_2 \in \{0, 1\}$), then Assumption 1 states that the change in loss possible due to changing $\phi(x_2)$ is higher than the loss difference (which is $0.1$ for $\ell_1$ loss). If $\mathcal{H}$ was the class of linear functions and we assume $\ell_1$ loss with all features in the same range (*e.g.*, $[0, 1]$), then Assumption 1 implies that the coefficient of the associational features in $f_{\theta_a}$ is higher than the change in coefficient for the causal features from $f_{\theta_c}$ to $f_{\theta_a}$.

---

[10]Note that $x_a$ and $x_c$ each represent a set of features, and not a single feature.

> **Lemma 2.** *Assume an LM adversary and a strongly convex loss function $\mathcal{L}$. Given a causal $f_{\theta_c}$ and an associational model $f_{\theta_a}$ trained on dataset $D$ using ERM. The LM adversary selects two points: $\mathbf{x}'$ and $\mathbf{x}''$. Then under Assumption 2, the* worst-case loss *obtained on the causal ERM model $\mathcal{L}_{\mathbf{x}'}(f_{\theta_c})$ is lower than the worst-case loss obtained on the associational ERM model $\mathcal{L}_{\mathbf{x}''}(f_{\theta_a})$ i.e.,*
>
> $$\mathcal{L}_{\mathbf{x}'}(f_{\theta_c}) \leq \mathcal{L}_{\mathbf{x}''}(f_{\theta_a})$$
>
> *which can be re-written as*
>
> $$\max_{\mathbf{x}} \mathcal{L}_{\mathbf{x}}(f_{\theta_c}) \leq \max_{\mathbf{x}} \mathcal{L}_{\mathbf{x}}(f_{\theta_a}) \tag{27}$$

**Proof**: Before we discuss the proof, let us establish another preliminary.

**Preliminary 2.** Let us write $f_{\theta_a}(\mathbf{x}) = f_{\theta_a}(x_c \cup x_a)$ as a combination of terms due to $x_c$ and $x_a$, where $x_c$ and $x_a$ are the causal features (parents) and non-causal features respectively *i.e.*, $x_c \cup x_a = \mathbf{x}$, and $x_c \cap x_a = \phi$. Let $\mathbf{x}' = x_c' \cup x_a'$ be the point chosen by the causal adversary.

We will show that the associational adversary can always choose a point $\mathbf{x}'' = x_c' \cup x_a''$ such that loss of the adversary is higher. We write, for any value $x_a$[11],

$$
\begin{aligned}
\mathcal{L}(f_{\theta_a}(x_c' \cup x_a'')) &= \mathcal{L}(f_{\theta_a}(x_c' \cup x_a'')) - \mathcal{L}(f_{\theta_a}(x_c' \cup x_a)) + \mathcal{L}(f_{\theta_a}(x_c' \cup x_a)) \\
&= (\mathcal{L}(f_{\theta_a}(x_c' \cup x_a'')) - \mathcal{L}(f_{\theta_a}(x_c' \cup x_a))) + (\mathcal{L}(f_{\theta_a}(x_c' \cup x_a)) \\
&\quad - \mathcal{L}(f_{\theta_c}(x_c' \cup x_a))) + \mathcal{L}(f_{\theta_c}(x_c' \cup x_a))
\end{aligned} \tag{28}
$$

Rearranging terms, and since $\mathcal{L}(f_{\theta_c}(x_c' \cup x_a')) = \mathcal{L}(f_{\theta_c}(x_c' \cup x_a))$ for any value of $x_a$ (causal model does not depend on associational features),

$$
\mathcal{L}(f_{\theta_a}(x_c' \cup x_a'')) - \mathcal{L}(f_{\theta_c}(x_c' \cup x_a')) = \underbrace{(\mathcal{L}(f_{\theta_a}(x_c' \cup x_a'')) - \mathcal{L}(f_{\theta_a}(x_c' \cup x_a)))}_{\text{Term 1}} \\
- \underbrace{(\mathcal{L}(f_{\theta_c}(x_c' \cup x_a)) - \mathcal{L}(f_{\theta_a}(x_c' \cup x_a)))}_{\text{Term 2}} \tag{29}
$$

Now the first term is $\geq 0$ since the adversary can select $x_a''$ such that loss increases (or stays constant) for $f_{\theta_a}$. Since the true function $f^*$ does not depend on $x_a$, changing $x_a$ does not change the true function's value but will change the value of the associational model (and adversary can choose it such that loss on the new point is higher). The second term can either be positive or negative. If it is negative, then we are done. Then the LHS $> 0$.

If the second term is positive, then we need to show that the first term is higher in magnitude than the second term. From assumption 1, let it be satisfied for some $x_a^\circ$. We know that $\mathcal{L}(f_{\theta_c}(x_c' \cup x_a')) = \mathcal{L}(f_{\theta_c}(x_c' \cup x_a^\circ))$ since the causal model ignores the associational features.

$$
\mathcal{L}(f_{\theta_c}(x_c' \cup x_a^\circ)) - \mathcal{L}(f_{\theta_a}(x_c' \cup x_a^\circ)) \leq \max_{x_c}(\mathcal{L}(f_{\theta_c}(x_c \cup x_a^\circ)) - \mathcal{L}(f_{\theta_a}(x_c \cup x_a^\circ)))
$$
$$
\leq \min_{x_c} \max_{x_a^*}(\mathcal{L}(f_{\theta_a}(x_c \cup x_a^*) - \mathcal{L}(f_{\theta_a}(x_c \cup x_a^\circ)) \leq \max_{x_a^*}(\mathcal{L}(f_{\theta_a}(x_c' \cup x_a^*) - \mathcal{L}(f_{\theta_a}(x_c' \cup x_a^\circ)) \tag{30}
$$

Now suppose adversary chooses a point such that $x_a'' = x_a^{max}$ where $x_a^{max}$ is the arg max of the RHS above. Then Equation 7 can be rewritten as

$$
\begin{aligned}
\mathcal{L}(f_{\theta_a}(x_c' \cup x_a^{max})) - \mathcal{L}(f_{\theta_c}(x_c' \cup x_a')) &= (\mathcal{L}(f_{\theta_a}(x_c' \cup x_a^{max})) - \mathcal{L}(f_{\theta_a}(x_c' \cup x_a^\circ))) \\
&\quad - (\mathcal{L}(f_{\theta_c}(x_c' \cup x_a^\circ)) - \mathcal{L}(f_{\theta_a}(x_c' \cup x_a^\circ))) \\
&> 0
\end{aligned} \tag{31}
$$

where the last inequality is due to equation 8.

Thus, adversary can always select a different value of $\mathbf{x} = x_c' \cup x_a^{max}$ such that loss is higher than the max loss in a causal model.

$$
\mathcal{L}(f_{\theta_c}(x_c' \cup x_a')) = \max_{\mathbf{x}} \mathcal{L}_{\mathbf{x}}(f_{\theta_c}) \leq \mathcal{L}(f_{\theta_a}(x_c' \cup x_a^{max})) \leq \max_{\mathbf{x}} \mathcal{L}_{\mathbf{x}}(f_{\theta_a})
$$

---

[11]We omit the subscript for $\mathcal{L}$ for brevity. It can be implied from context.

B.5    PROOF OF STEP 3 FROM OUTLINE

> **Theorem 2.** *Assume the existence of a dataset $D$ of $n$ samples. Further, assume a neighboring dataset is defined by adding a data point to $D$. Let $f_{\theta_c}$ and $f_{\theta_a}$ be the causal and associational models learnt using $D$, and $f_{\theta'_c}$ and $f_{\theta'_a}$ be the causal and associational models learnt using neighboring datasets $D'$ and $D''$ respectively. All models are obtained by ERM on a Lipschitz continuous (with parameter $\rho$), strongly convex (with parameter $\lambda$) loss function $\mathcal{L}$. Then, the sensitivity of a* causal *learning function $H_C$ will be lower than that of its associational counterpart $H_A$. Mathematically, assuming large enough $n$ such that $n > \frac{2\rho}{\lambda} - 1$,*
>
> $$\max_{D,D'} ||\theta_c - \theta'_c|| \leq \max_{D,D'} ||\theta_a - \theta'_a|| \tag{32}$$

**Proof:** The proof uses strongly convex and Lipschitz properties of the loss function. Before we discuss the proof, let us introduce a requisite preliminary.

**Preliminary 3.** Assume the existence of a dataset $D$ of size $n$. There are two generative models learnt, $f_{\theta_a}$ and $f_{\theta_c}$ using this dataset. Similarly, assume there is a neighboring dataset $D'$ which is obtained by adding one point $\mathbf{x}'$. Then the corresponding ERM models learnt using $D'$ are $f_{\theta'_a}$ and $f_{\theta'_c}$.

We now detail the steps of the proof.

**Step 1.** Assume $\mathcal{L}$ is strongly convex. Then by the optimality of ERM predictor on $D$ and the definition of strong convexity,

$$
\begin{aligned}
\mathcal{L}_D(f_\theta) &\leq \mathcal{L}_D(\alpha f_\theta + (1-\alpha)f_{\theta'}) \\
&\leq \alpha \mathcal{L}_D(f_\theta) + (1-\alpha)\mathcal{L}_D(f_{\theta'}) - \frac{\lambda}{2}\alpha(1-\alpha)||\theta - \theta'||^2
\end{aligned}
\tag{33}
$$

**Step 2.** Rearranging terms, and as $\alpha \to 1$,

$$
\begin{aligned}
(1-\alpha)\mathcal{L}_D(f_\theta) &\leq (1-\alpha)\mathcal{L}_D(f_{\theta'}) - \frac{\lambda}{2}\alpha(1-\alpha)||\theta - \theta'||^2 \\
&\Rightarrow ||\theta - \theta'||^2 \leq \frac{2}{\lambda}(\mathcal{L}_D(f_{\theta'}) - \mathcal{L}_D(f_\theta))
\end{aligned}
\tag{34}
$$

**Step 3.** Further, we can write $(\mathcal{L}_D(f_{\theta'}) - \mathcal{L}_D(f_\theta))$ in terms of loss on $\mathbf{x}'$.

$$
\begin{aligned}
\mathcal{L}_{D'}(f_\theta) &= \frac{n}{n+1}\mathcal{L}_D(f_\theta) + \frac{1}{n+1}\mathcal{L}_{\mathbf{x}'}(f_\theta) \text{ (since } D' = D \cup \mathbf{x}') \\
&\leq \frac{n}{n+1}\mathcal{L}_D(f_{\theta'}) + \frac{1}{n+1}\mathcal{L}_{\mathbf{x}'}(f_\theta) \text{ (from Equation 34)} \\
&\leq \frac{n}{n+1}\frac{n+1}{n}\mathcal{L}_{D'}(f_{\theta'}) - \frac{n}{n+1}\frac{1}{n}\mathcal{L}_{\mathbf{x}'}(f_{\theta'}) + \frac{1}{n+1}\mathcal{L}_{\mathbf{x}'}(f_\theta) \text{ (since } D = D' - \{\mathbf{x}'\}) \\
\Rightarrow \mathcal{L}_{D'}(f_\theta) - \mathcal{L}_{D'}(f_{\theta'}) &\leq \frac{1}{n+1}(\mathcal{L}_{\mathbf{x}'}(f_\theta) - \mathcal{L}_{\mathbf{x}'}(f_{\theta'}))
\end{aligned}
\tag{35}
$$

**Step 4.** Combining the above two equations, we obtain,

$$||\theta - \theta'||^2 \leq \frac{2}{\lambda}(\mathcal{L}_D(f_{\theta'}) - \mathcal{L}_D(f_\theta)) \leq \frac{2}{\lambda(n+1)}(\mathcal{L}_{\mathbf{x}'}(f_\theta) - \mathcal{L}_{\mathbf{x}'}(f_{\theta'})) \tag{36}$$

**Step 5.** From Claim 1 above, we know that

$$
\begin{aligned}
\max_{\mathbf{x}} \mathcal{L}_{\mathbf{x}}(f_{\theta_c}) &\leq \max_{\mathbf{x}} \mathcal{L}_{\mathbf{x}}(f_{\theta_a}) \\
&\Rightarrow \mathcal{L}_{\mathbf{x}'}(f_{\theta_c}) \leq \mathcal{L}_{\mathbf{x}''}(f_{\theta_a})
\end{aligned}
\tag{37}
$$

where $\mathbf{x}' = \arg\max_{\mathbf{x}} \mathcal{L}_{\mathbf{x}}(f_{\theta_c})$ and $\mathbf{x}''$ is chosen such that $\mathbf{x}'$ and $\mathbf{x}''$ differ only in the associational features. Thus, $\mathcal{L}_{\mathbf{x}'}(f_{\theta'_c}) = \mathcal{L}_{\mathbf{x}''}(f_{\theta'_c})$. Also because $\mathcal{H}_C \subseteq \mathcal{H}_A$, the training loss of the ERM model for any $D''$ defined using $D$ and $\mathbf{x}''$ is higher for a causal model *i.e.,*

$$\mathcal{L}_{\mathbf{x}'}(f_{\theta'_c}) = \mathcal{L}_{\mathbf{x}''}(f_{\theta'_c}) \geq \mathcal{L}_{\mathbf{x}''}(f_{\theta'_a}) \tag{38}$$

Therefore, we obtain,

$$\mathcal{L}_{\mathbf{x}'}(f_{\theta_c}) - \mathcal{L}_{\mathbf{x}'}(f_{\theta'_c}) = \max_{\mathbf{x}} \mathcal{L}_{\mathbf{x}}(f_{\theta_c}) - \mathcal{L}_{\mathbf{x}'}(f_{\theta'_c}) \leq \mathcal{L}_{\mathbf{x}''}(f_{\theta_a}) - \mathcal{L}_{\mathbf{x}''}(f_{\theta'_a}) \tag{39}$$

So we have now shown that the max loss difference on a point $\mathbf{x}'$ for causal ERM models trained on neighboring datasets is lower than the corresponding loss difference over $\mathbf{x}''$ for the associational models.

**Step 6.** Now we use the Lipschitz property, to claim,

$$\mathcal{L}_{\mathbf{x}''}(f_{\theta_a}) - \mathcal{L}_{\mathbf{x}''}(f_{\theta'_a}) \leq \rho ||\theta_a - \theta'_a|| \tag{40}$$

**Step 7.** Combining Equations 36 (substituting $f_{\theta_c}$) and 40, and taking max on the RHS, we get,

$$\max_{D,D'} ||\theta_c - \theta'_c||^2 \leq \frac{2}{\lambda(n+1)} \max_{D,D'} \mathcal{L}_{\mathbf{x}'}(f_{\theta_c}) - \mathcal{L}_{\mathbf{x}'}(f_{\theta'_c})$$

$$\leq \frac{2}{\lambda(n+1)} \max_{D,D'} \mathcal{L}_{\mathbf{x}'}(f_{\theta_a}) - \mathcal{L}_{\mathbf{x}'}(f_{\theta'_a}) \tag{41}$$

$$\leq \frac{2\rho}{\lambda(n+1)} \max_{D,D'} ||\theta_a - \theta'_a||$$

$$\Rightarrow \max_{D,D'} ||\theta_c - \theta'_c||^2 \leq \frac{2\rho}{\lambda(n+1)} \max_{D,D'} ||\theta_a - \theta'_a|| \tag{42}$$

For $n + 1 > \frac{2\rho}{\lambda}$,

$$\max_{D,D'} ||\theta_c - \theta'_c||^2 \leq \max_{D,D'} ||\theta_a - \theta'_a|| \tag{43}$$

Now if $\max_{D,D'} ||\theta_c - \theta'_c|| \geq 1$, then the result follows by taking the square root over LHS. If not, we need a sufficiently large $n$ such that $n + 1 > \frac{2\rho}{\lambda \max_{D,D'} ||\theta_c - \theta'_c||}$, then we obtain,

$$\max_{D,D'} ||\theta_c - \theta'_c|| \leq \max_{D,D'} ||\theta_a - \theta'_a|| \tag{44}$$

**Remark on Theorem 2.** Theorem 2 depends on two key assumptions:

1. Assumption 1 that constrains associational model to have non-trivial contribution from associational (non-causal) features.
2. A sufficiently large $n$ as shown above.

When any of these assumptions is violated (*e.g.,* a small-$n$ training dataset or an associational model that is negligibly dependent on the associational features), then it is possible that the causal ERM model has higher $\varepsilon$ than the associational model.

> **Connections between Theory & Practice:** The proof provided in Appendix A is meant to provide intuition as to the benefit of causal information in a simpler setting using linear gaussian mixtures, similar to what is done in other work (Ilyas et al., 2019). We provide a more general proof (across all classes of generative models), for a general privacy adversary in Appendix B, but under some constraining assumptions regarding the convexity of the loss function.

## C  TRAINING DETAILS

### C.1  OVERALL SCHEME

We outline the work-flow overview (refer Figure 7): 1. The user provides a dataset for which they want to create a synthetic copy; 2. We utilize techniques (e.g., Morales-Alvarez et al. (2021)) to learn the structured causal model (SCM) associated with this dataset (or can assume the SCM is given); 3. We encode this structure into a generative model (e.g., Morales-Alvarez et al. (2021); Geffner et al. (2022); Kyono et al. (2021)) and train it with DP-SGD; 4. The generative model is sampled to obtain a synthetic data which is DP (by post-processing) and can be used for arbitrary downstream tasks. We will also clarify that unlike some works which require some information about the nature of usage of downstream data, ours does not.

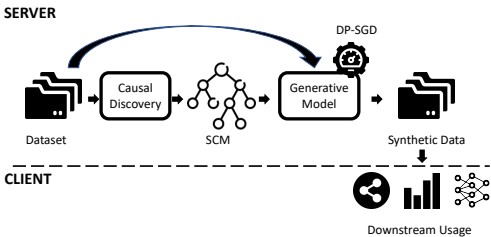

Figure 7: Overview of procedure.

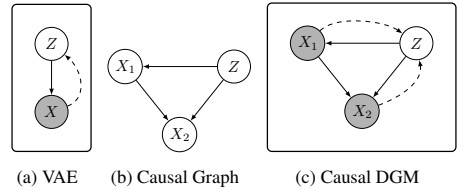

(a) VAE     (b) Causal Graph     (c) Causal DGM

Figure 9: Exemplar case comparing our solution to VAE.

### C.2  MODELS

**Pain**

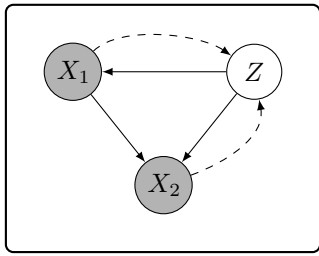

**Encoder:** $q_\phi(z, x_1 | x_2) = q_{\phi_1}(z | x_1, x_2) \cdot q_{\phi_2}(x_1 | x_2)$

**Decoder:** $p_\theta(x_2, x_1, z) = p_{\theta_1}(x_2 | x_1) \cdot p_{\theta_2}(x_2 | z) \cdot p_{\theta_3}(x_1 | z)$

**EEDI**

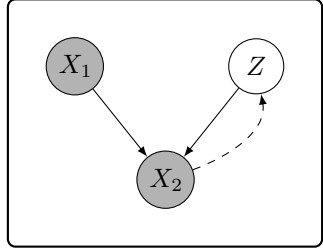

**Encoder:** $q_\phi(z, x_1 | x_2) = q_{\phi_1}(z | x_2)$

**Decoder:** $p_\theta(x_2, x_1, z) = p(z) \cdot p(x_1) \cdot p_{\theta_1}(x_2 | z) \cdot p_{\theta_2}(x_2 | x_1)$

**Note:** All encoders and decoders used as part of our experiments comprised of simple feed-forward architectures. In particular, these architectures have 3 layers. All embeddings generated are of size 10. We set the learning rate to be 0.001.

### C.3 DATASETS

| Dataset | # Records | $k$ | $\varepsilon$ |
|---|---|---|---|
| EEDI Wang et al. (2020b) | 2950 | 948 | 12.42 |
| Pain5000 Tu et al. (2019) | 5000 | 222 | 5.62 |
| Pain1000 Tu et al. (2019) | 1000 | 222 | 2.36 |
| Synthetic | 1000 | 22 | 3.9 |
| Lung Cancer Lauritzen & Spiegelhalter (1988) | 80,000 | 8 | - |

Table 2: Salient features of our experimental setup. More information about the datasets and parameters used can be found in the Appendix C.4.

We evaluate on datasets from three real-world applications. The first one is the `EEDI` dataset Wang et al. (2020b) which is one of the largest real-world education data collected from an online education platform. It contains answers by students (of various educational backgrounds) for certain diagnostic questions. The second one is the neuropathic pain (`Pain`) diagnosis dataset obtained from a causally grounded simulator Tu et al. (2019). For this dataset, we consider two variants: one with 1000 data records (or `Pain1000`), and another with 5000 data records (or `Pain5000`). The third dataset (`Lung Cancer`) contains information about lung diseases and visits to Asia Lauritzen & Spiegelhalter (1988).

More salient features of each dataset is presented in Table 2. We choose these datasets as they encompass diversity in their size ($n$), dimensionality ($k$), and have some prior information on causal structures (refer Appendix D). We utilize this causal information in building a causally informed generative models. The (partial) SCM is given utilizing domain knowledge of the `EEDI`[12] and `Pain` contexts.

---

[12]A larger SCM for the `EEDI` dataset was learnt using the VICause methodology proposed by Morales-Alvarez et al. (2021).

### C.4 TRAINING PARAMETERS

| Causality | Dataset | Batch Size | Epochs |
|:---:|:---:|:---:|:---:|
| $\times$ | EEDI | 200 | 1000 |
| $\checkmark$ | EEDI | 200 | 1000 |
| $\times$ | Pain1000 | 100 | 100 |
| $\checkmark$ | Pain1000 | 100 | 100 |
| $\times$ | Pain5000 | 100 | 100 |
| $\checkmark$ | Pain5000 | 100 | 100 |
| $\times$ | Synthetic | 100 | 50 |
| $\checkmark$ | Synthetic | 100 | 50 |
| $\checkmark$ | Lung Cancer | 100 | 100 |

Table 3: Training parameters for our experimental evaluation.

For the experiments related to the model proposed by Morales-Alvarez et al. (2021), we utilized the same parameters as for EEDI (in Table 3) to obtain the same privacy expenditure ($\varepsilon$).

### D CAUSAL GRAPHS

**SCM-1:** SCM for the Pain dataset. $X_1$ denotes the causes of a medical condition, and $X_2$ denotes the various conditions.

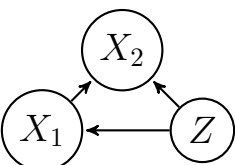

**SCM-2:** SCM used by the EEDI. $X_2$ denotes the answers to questions and $X_1$ is the student meta data such as the year group and school.

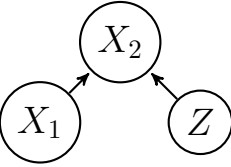

**VICause:** We refer the reader to the work of Morales-Alvarez Morales-Alvarez et al. (2021), which contains information about the causal graph used to obtain Figure 2.

**Lung Cancer:** The causal graph related to the Lung Cancer dataset can be found in `https://www.bnlearn.com/bnrepository/discrete-small.html`. The edge between `tub` and `ether` was removed to simulate missing edges, and the edge between `asia` and `smoke` was added to simulate new edges.

# E  UTILITY EVALUATION

## E.1  ACCURACY ON ORIGINAL DATA

The results are detailed in Table 4.

| Dataset | Non Causal | | | | | Causal | | | | |
|---|---|---|---|---|---|---|---|---|---|---|
| | kernel | svc | logistic | rf | knn | kernel | svc | logistic | rf | knn |
| EEDI | 86.93 | 91.24 | 88.53 | 93.8 | 88.22 | 87.38 | 91.46 | 88.95 | 93.44 | 87.92 |
| Pain1000 | 92.75 | 94.42 | 91.56 | 94.19 | 88.78 | 92.69 | 94.03 | 91.81 | 94.28 | 89.5 |
| Pain5000 | 95.37 | 96.53 | 97.47 | 93.03 | 92.14 | 95.48 | 96.5 | 97.39 | 92.74 | 92.19 |

Table 4: **Baseline Accuracy** calculated on the original (and not synthetic) data. Results presented in Table 1 are based on these values.

## E.2  PAIRPLOTS

Observe that the pair-plots obtained from the models trained with causality and DP are comparable to those obtained from models trained with causality and no DP; the utility of both these models should be comparable.

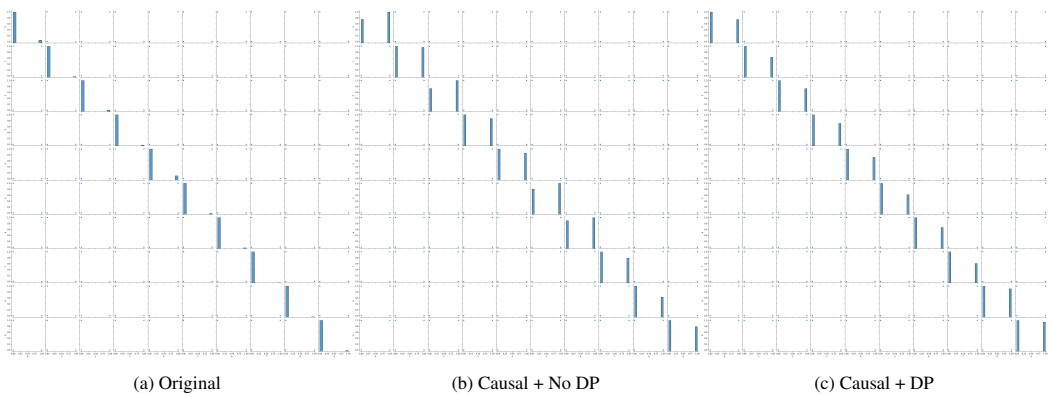

(a) Original      (b) Causal + No DP      (c) Causal + DP

Figure 10: `Pain5000` Dataset

# F  RESULTS ON PAIN1000 DATASET

As in the figures in the main body of the paper, we plot the average success of the MI attack Stadler et al. (2020). From Figures 11 and 12, observe that the trends are the same as what we observed earlier: causality in conjunction with DP reduces the advantage of the adversary. DP and causality in isolation also reduce the MI adversary's advantage.

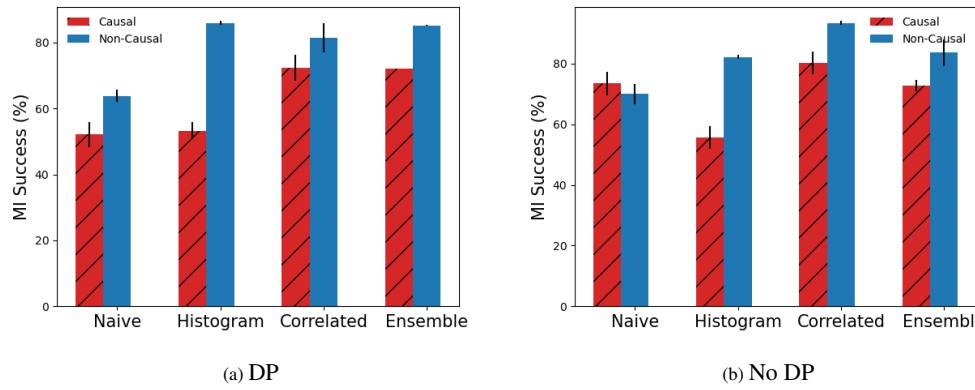

Figure 11: **Impact of DP on MI resilience:** Causality by itself is able to provide resilience to MI adversaries. When combined with DP, the advantage is exacerbated.

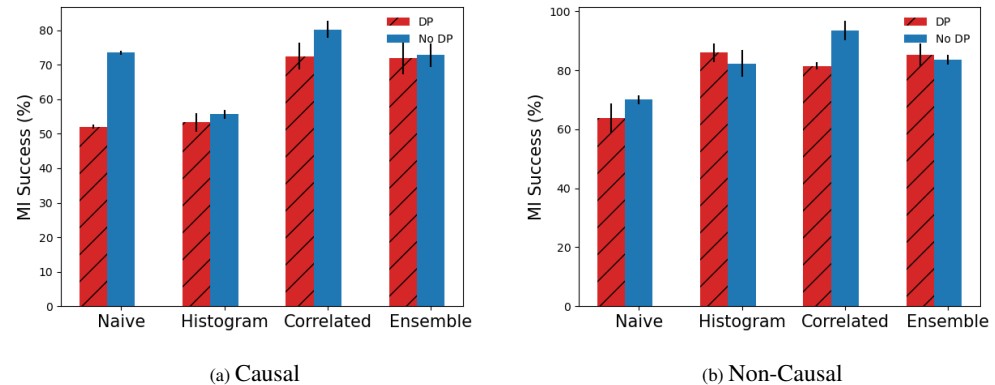

Figure 12: **Impact of Causality on MI resilience:** Observe that the combination of DP and causality produces more resilience to MI attacks.

# G MI RESULTS

**PA** denotes the ability of the classifier to correctly classify train samples. **NA** denotes the ability of the classifier to correctly classify test samples. The **Accuracy** is a weighted combination of **PA** and **NA**.

**EEDI.** Observe that when there is causal information, the difference between the DP and No DP column is larger.

| Extractor | Attack Model | Accuracy | | NA | | PA | |
|---|---|---|---|---|---|---|---|
| | | DP | No DP | DP | No DP | DP | No DP |
| Naive | kernel | 47.33 | 47.33 | 100 | 100 | 0 | 0 |
| | svc | 47.33 | 47.33 | 100 | 100 | 0 | 0 |
| | random forest | 62.18 | 99.21 | 58.51 | 99.49 | 65.48 | 98.96 |
| | knn | 54.79 | 96.48 | 55.19 | 98.85 | 54.43 | 94.36 |
| Histogram | kernel | 100 | 100 | 100 | 100 | 100 | 100 |
| | svc | 100 | 100 | 100 | 100 | 100 | 100 |
| | random forest | 100 | 100 | 100 | 100 | 100 | 100 |
| | knn | 100 | 100 | 100 | 100 | 100 | 100 |

Table 5: Model trained using `EEDI` and partial causal information.

| Extractor | Attack Model | Accuracy | | NA | | PA | |
|---|---|---|---|---|---|---|---|
| | | DP | No DP | DP | No DP | DP | No DP |
| Naive | kernel | 49.85 | 81.23 | 100 | 100 | 0 | 62.58 |
| | svc | 75.38 | 81.23 | 62.96 | 100 | 87.73 | 62.58 |
| | random forest | 98.15 | 96.92 | 98.77 | 97.53 | 97.55 | 96.32 |
| | knn | 98.15 | 94.46 | 99.38 | 95.06 | 96.93 | 93.87 |
| Histogram | kernel | 100 | 100 | 100 | 100 | 100 | 100 |
| | svc | 100 | 100 | 100 | 100 | 100 | 100 |
| | random forest | 100 | 100 | 100 | 100 | 100 | 100 |
| | knn | 100 | 100 | 100 | 100 | 100 | 100 |

Table 6: Model trained using `EEDI` and causal information obtained from VICause Morales-Alvarez et al. (2021).

| Extractor | Attack Model | Accuracy | | NA | | PA | |
|---|---|---|---|---|---|---|---|
| | | DP | No DP | DP | No DP | DP | No DP |
| Naive | kernel | 47.33 | 47.33 | 100 | 100 | 0 | 0 |
| | svc | 53.21 | 60.85 | 60.44 | 57.62 | 46.72 | 63.75 |
| | random forest | 99.76 | 100 | 99.74 | 100 | 99.77 | 100 |
| | knn | 99.76 | 99.82 | 99.87 | 99.62 | 99.65 | 100 |
| Histogram | kernel | 52.67 | 100 | 0 | 100 | 100 | 100 |
| | svc | 47.33 | 95.7 | 100 | 95.39 | 0 | 95.97 |
| | random forest | 59.39 | 100 | 100 | 100 | 22.9 | 100 |
| | knn | 59.39 | 100 | 100 | 100 | 22.9 | 100 |

Table 7: Model trained using `EEDI` and no causal information.

**Pain 1000.** Observe that when there is causal information, the difference between the DP and No DP column is larger.

| Extractor | Attack Model | Accuracy | | NA | | PA | |
|---|---|---|---|---|---|---|---|
| | | DP | No DP | DP | No DP | DP | No DP |
| Naive | kernel | 47.33 | 47.33 | 100 | 100 | 0 | 0 |
| | svc | 47.33 | 47.33 | 100 | 100 | 0 | 0 |
| | random forest | 57.7 | 99.58 | 60.44 | 99.74 | 55.24 | 99.42 |
| | knn | 55.94 | 96.76 | 60.44 | 96.16 | 51.9 | 95.4 |
| Histogram | kernel | 48.3 | 57.82 | 78.36 | 100 | 21.29 | 19.91 |
| | svc | 47.33 | 47.33 | 100 | 100 | 0 | 0 |
| | random forest | 57.82 | 57.82 | 100 | 100 | 19.91 | 19.91 |
| | knn | 59.7 | 59.7 | 81.82 | 81.82 | 39.82 | 39.82 |
| Correlated | kernel | 99.7 | 98.24 | 100 | 99.62 | 99.42 | 97.01 |
| | svc | 51.09 | 62.12 | 83.99 | 50.58 | 21.52 | 72.5 |
| | random forest | 97.39 | 100 | 95.52 | 100 | 99.08 | 100 |
| | knn | 41.52 | 60.61 | 42.64 | 60.95 | 40.51 | 60.3 |
| Ensemble | kernel | 61.76 | 61.7 | 20.49 | 72.23 | 98.85 | 51.78 |
| | svc | 47.33 | 47.33 | 100 | 100 | 0 | 0 |
| | random forest | 97.94 | 100 | 96.93 | 100 | 98.85 | 100 |
| | knn | 81.09 | 82.12 | 86.3 | 83.99 | 76.41 | 80.44 |

Table 8: Model trained using `Pain1000` and partial causal information.

| Extractor | Attack Model | Accuracy | | NA | | PA | |
|---|---|---|---|---|---|---|---|
| | | DP | No DP | DP | No DP | DP | No DP |
| Naive | kernel | 47.33 | 47.33 | 100 | 100 | 0 | 0 |
| | svc | 47.33 | 47.33 | 100 | 100 | 0 | 0 |
| | random forest | 79.82 | 99.7 | 82.33 | 100 | 77.56 | 99.42 |
| | knn | 80.85 | 85.76 | 82.71 | 88.6 | 79.17 | 83.2 |
| Histogram | kernel | 96.36 | 96.79 | 96.41 | 90.14 | 96.32 | 83.77 |
| | svc | 47.33 | 47.33 | 100 | 100 | 0 | 0 |
| | random forest | 100 | 100 | 100 | 100 | 100 | 100 |
| | knn | 100 | 94.91 | 100 | 94.88 | 100 | 94.94 |
| Correlated | kernel | 100 | 100 | 100 | 100 | 100 | 100 |
| | svc | 94.12 | 100 | 92.83 | 100 | 95.28 | 100 |
| | random forest | 78 | 100 | 79 | 100 | 77.1 | 100 |
| | knn | 53.82 | 73.88 | 74.9 | 72.73 | 34.87 | 74.91 |
| Ensemble | kernel | 98.55 | 98.3 | 98.46 | 99.49 | 96.82 | 97.24 |
| | svc | 47.33 | 47.33 | 100 | 100 | 0 | 0 |
| | random forest | 94.85 | 100 | 93.98 | 100 | 95.63 | 100 |
| | knn | 100 | 88.79 | 100 | 90.27 | 100 | 87.46 |

Table 9: Model trained using `Pain1000` and no causal information.

**Pain 5000.** Observe that when there is causal information, the difference between the DP and No DP column is larger.

| Extractor | Attack Model | Accuracy | | NA | | PA | |
|---|---|---|---|---|---|---|---|
| | | DP | No DP | DP | No DP | DP | No DP |
| Naive | kernel | 47.33 | 47.33 | 100 | 100 | 0 | 0 |
| | svc | 47.33 | 47.33 | 100 | 100 | 0 | 0 |
| | random forest | 76.85 | 98.85 | 78.36 | 97.7 | 75.49 | 99.88 |
| | knn | 76.79 | 96.12 | 78.36 | 93.98 | 75.37 | 98.04 |
| Histogram | kernel | 47.33 | 60.24 | 100 | 38.16 | 0 | 80.09 |
| | svc | 47.33 | 47.33 | 100 | 100 | 0 | 0 |
| | random forest | 57.82 | 57.82 | 100 | 100 | 19.91 | 19.91 |
| | knn | 58.7 | 59.7 | 81.82 | 81.82 | 39.82 | 38.82 |
| Correlated | kernel | 92.61 | 100 | 94.88 | 100 | 90.56 | 100 |
| | svc | 47.33 | 97.64 | 100 | 100 | 0 | 95.51 |
| | random forest | 100 | 100 | 100 | 100 | 100 | 100 |
| | knn | 36.48 | 100 | 45.07 | 100 | 28.77 | 100 |
| Ensemble | kernel | 47.33 | 62.18 | 100 | 46.73 | 0 | 76.06 |
| | svc | 47.33 | 47.33 | 100 | 100 | 0 | 0 |
| | random forest | 100 | 100 | 100 | 100 | 100 | 100 |
| | knn | 74.85 | 100 | 81.56 | 100 | 68.81 | 100 |

Table 10: Model trained using `Pain5000` and partial causal information.

| Extractor | Attack Model | Accuracy | | NA | | PA | |
|---|---|---|---|---|---|---|---|
| | | DP | No DP | DP | No DP | DP | No DP |
| Naive | kernel | 47.33 | 47.33 | 100 | 100 | 0 | 0 |
| | svc | 64.61 | 47.33 | 60.18 | 100 | 68.58 | 0 |
| | random forest | 99.94 | 100 | 99.87 | 100 | 100 | 100 |
| | knn | 100 | 99.82 | 100 | 99.62 | 100 | 100 |
| Histogram | kernel | 100 | 100 | 100 | 100 | 100 | 100 |
| | svc | 65.03 | 47.33 | 91.55 | 100 | 41.2 | 0 |
| | random forest | 100 | 100 | 100 | 100 | 100 | 100 |
| | knn | 100 | 100 | 100 | 100 | 100 | 100 |
| Correlated | kernel | 100 | 100 | 100 | 100 | 100 | 100 |
| | svc | 100 | 100 | 100 | 100 | 100 | 100 |
| | random forest | 99.76 | 100 | 99.62 | 100 | 99.88 | 100 |
| | knn | 53.09 | 100 | 80.41 | 100 | 28.54 | 100 |
| Ensemble | kernel | 100 | 100 | 100 | 100 | 100 | 100 |
| | svc | 48.85 | 47.33 | 100 | 100 | 0 | 0 |
| | random forest | 100 | 100 | 100 | 100 | 100 | 100 |
| | knn | 100 | 100 | 100 | 100 | 100 | 100 |

Table 11: Model trained using `Pain5000` and no causal information.

