# OpenReview forum: "Causally Constrained Data Synthesis For Private Data Release"
_ICLR.cc/2023/Conference — Submitted to ICLR 2023_

### Official Review · Reviewer_2NXW · 2022-10-21

**Confidence:** 3
**Correctness:** 2
**Technical Novelty And Significance:** 3
**Empirical Novelty And Significance:** 2
**Recommendation:** 3

**Clarity, Quality, Novelty And Reproducibility:**

Causality is central to the claim made by the paper, but it is not clear from the results (theoretical or empirical) than the improved privacy/utility tradeoff is due to causal information or rather from just being able to factorize the distribution correctly / having a sparser model. The paper even indicates at the end of page 4 that this is why performance is better, but then does not address whether performance would be the same or different if you simply used a different (incorrect) causal model from the same Markov equivalence class. Thus, it is not clear the claims about causality are correct.

If the impact is due not to causality, but rather having a sparser model, then the results are maybe not surprising (you’ve simply eliminated some unassociated variables that would add noise) and hence can get a better privacy/utility tradeoff. In most cases, however, you would not have this sparse model for free, but would need to learn it and dedicate some of your privacy budget to learning the correct Markov equivalence class. The paper discusses this, but most of the empirical results assume some knowledge of the underlying graph.

The empirical results also leave some questions unanswered. In many cases using the synthetic data (even in some cases with DP) actually consistently increases performance on the downstream task even over the real data, which is surprising since the generative model is not doing any augmentation aimed at increasing performance on the specific task. This is not explained in the text and the performance differences are listed without confidence intervals so it’s unclear how robust these results are.

**Strength And Weaknesses:**

Strengths
- The paper address a topic of relevance to multiple research communities
- The paper includes a nice mix of theory and empirical results
- The paper is generally well written

Weaknesses
- There appear to be gaps between the claims made in the paper and the results which makes the significance unclear (see below)
- Some results are not well explained
- Some assumptions may not align to realistic settings
- It is not clear how robust some of the empirical results are

**Summary Of The Paper:**

The paper considers the effect of adding causality to differentially private generative models. A theorem is proved showing that the sensitivity of a causally informed mechanism is lower than a non-causal mechanism. Experiments show that using causal information with the same privacy budget results in models which perform better on downstream tasks than generative models which do not use causal information.

**Summary Of The Review:**

The paper addresses an interesting problem and includes a nice mix of theoretical and empirical results, but there appear to be some gaps between the theory and claims and the robustness of the empirical results are unclear.

---

> ### Author Response · Authors · 2022-11-15
> **Thank you for your feedback!**
>
> Response to comment on correctness of causal graph: We would like to point out that the claim we make at the end of page 4 is that causal learning enables learning of the “correct factorization”. We do not state that this results in a model that is sparser; not in page 4 or anywhere else in the paper. All graphs within the Markov equivalence class are “correct”, so we are having a difficult time understanding the reviewer's claims. We do state in the introduction that we evaluate the efficacy of an incorrect causal graph on the overall procedure; the same is presented in Section 5.2.2 and 5.2.3.
>
> Response to model sparsity: Our particular implementation involves an associational model with“140738” parameters, and a causal model with “268428” parameters (for the EEDI dataset). Thus, causality does not induce a sparser model, but instead introduces an architecture where the connections between neurons are different (and faithful to the causal graph). We agree with the reviewer’s comment about learning the causal graph and this would require some part of the privacy budget as well, but as we state in the introduction – such a goal is orthogonal to the ones in the paper (as prior work tackles it). Additionally, in our experiments with VICAUSE (i..e, technique by Morales-Alvarez et al.), learning the causal graph with DP is achievable by modifying the learning algorithm to incorporate DP-SGD as well.
>
> Response to presentation of results: The reviewer is correct in observing that there are no confidence intervals reported in Table 1. We also apologize for the ambiguity in our setup. The setup we consider is as follows: given a dataset with n records and k attributes, we wish to learn a function that predicts the ‘k’th attribute given access to k-1 other attributes. We vary the choice of the ‘k’th attribute in our experiments (20 random choices) – which results in the “range of baseline utility” as shown in the first column. We can repeat trials (i.e., run 5 trials for each choice of k) and report the range along with some confidence information. Then the numbers reported in the cells of Table 1 are the “average change in utility” across these 20 selections. We can report variance of these numbers as well.
>
> We hope you find these answers satisfactory. If so, please do consider increasing the score for the submission. If not, kindly let us know what better we can clarify.

---

### Official Review · Reviewer_3noA · 2022-10-25

**Confidence:** 2
**Correctness:** 3
**Technical Novelty And Significance:** 2
**Empirical Novelty And Significance:** 2
**Recommendation:** 3

**Clarity, Quality, Novelty And Reproducibility:**

Clarity: The clarity of this work can be improved. The datasets (page 8) and experiments (page 7) are not described until the nearly the end of the paper, which makes it very confusing to read the results on page 6. There were, however, some helpful parts, such as the description of the notation on page 3 (eg upper-case variables = sets, etc).

Quality: This paper seems to still be a bit of a work-in-progress. It shows promising directions, particularly w.r.t. evaluating privacy extrinsically without just reporting the privacy budget. However, the experiments and presented results are a bit scattered in terms of the picture that is trying to be painted.

Novelty: This work is moderately novel. It aims to use causal information to structure a differentiably private generative model. All of the techniques are reasonably well-used, but combining them has some degree of novelty.

Reproducibility: There is no code available. This work would likely be difficult to reproduce without additional resources and guidance.

**Strength And Weaknesses:**

STRENGTHS
- Rather than just relying on the privacy budget (epsilon) to quantify privacy, this work uses  susceptibility to a membership inference attack. It is very valuable to explore quantifiable notions of privacy based on risk (eg in order to compare privacy risks between DP-based methods and non-DP methods).

WEAKNESSES
- I am not persuaded that this was a good measure of "utility." Page 5 describes that the metric involves randomly selecting 20 attributes to be predicted, rather than picking actually useful targets. This is especially surprising given this work's emphasis on the importance of causal and meaningful relationships in data generation. I find the approach concerning because it is essentially performing an *attribute* inference attack (ie if I know someone's age, gender, etc, can I predict their HIV status using the synthetic data?). Without more details & careful thought, this metric could arguably be another measurement of risk/privacy instead of utility.
- On multiple occasions, the work explicitly stresses that its contribution is meant to be severable from the construction of a given causal graph. However, I'm not sure such a separation can be so clean. The field of causal inference suffers in reputation (and ultimately empirical performance) because although "the causal graph can be constructed using domain expertise" in theory... in practice, trying to construct a meaningful graph often leads to over-promise/under-deliver. The work could benefit from showing the CG for the 3-node experiment.

**Summary Of The Paper:**

This paper tries to measure whether adding causal information to a differentiably private generative model allows for a more favorable utility-privacy tradeoff curve.

**Summary Of The Review:**

This paper seems to still be a bit of a work-in-progress. It shows promising directions, particularly w.r.t. evaluating privacy extrinsically without just reporting the privacy budget. However, the experiments and presented results are a bit scattered in terms of the picture that is trying to be painted. The two largest areas of improvement I would suggest are: 1) either reworking the notion of utility or at least further justifying why it is appropriate; and 2) re-organizing some of the sections of writing so that the experiments flow more meaningfully and coherently.

---

> ### Author Response · Authors · 2022-11-15
> **Thank you for your feedback!**
>
> Response to weakness 1: We do not understand this comment for the following reasons: 1. We are not in a position to judge what is useful and what is not; 2. The evaluation shows what happens in the average case: such a study is also beneficial (for privacy and utility analysis) when compared to the proposed worst case measurement; 3. The exact same procedure is followed in prior work related to measuring the utility of DP synthetic data generation (e.g., http://dimacs.rutgers.edu/~graham/pubs/papers/PrivBayes.pdf, https://arxiv.org/abs/2012.15128 and https://openreview.net/forum?id=S1zk9iRqF7). Penalising us for making the same decisions as previously published (and well-cited papers) is highly non-standard and unfair.
>
> Response to weakness 2: The 3 node CG used is presented in Appendix C2. We definitely agree with the reviewer in spirit: the correctness of the causal graph learning mechanism has implications on the utility vs. empirical privacy of the approach (as discussed in Section 5.2.2). However, we believe that as these algorithms become better, the aforementioned empirical measurements will also be better. Our theory, on the other hand, discusses an “ideal situation” where the graph being learnt is truly faithful to the underlying data generating process. We show how, in such a situation, there is a privacy amplification induced (and this is disconnected from the practical implementation in the status quo). We urge the reviewer to consider this in their calibration of their score.
>
> Response to clarity: We thank the reviewer for valuable suggestions on how to improve the presentation of our paper. We will gladly incorporate them.
>
> Response to quality: Could the reviewer kindly elaborate upon this statement? This seems subjective, and there is nothing concrete in it for us to respond to.
>
> Response to novelty: We thank the reviewer for noting the novelty in our combination of techniques. We would also like to stress that apart from the empirical results we have (and the engineering efforts associated with getting causally consistent models to train with DP), we have new theoretical results that discuss the privacy amplification induced by causality. Such a result does not exist in prior work related to generative models, and is interesting in its own right.
>
> Response to reproducibility: The code used for the experiments is not licensed for release, but we will actively work on ensuring that it will be made available.
>
> We hope you find these answers satisfactory. If so, please do consider increasing the score for the submission. If not, kindly let us know what better we can clarify.

---

### Official Review · Reviewer_PkKv · 2022-10-26

**Confidence:** 4
**Correctness:** 3
**Technical Novelty And Significance:** 3
**Empirical Novelty And Significance:** 2
**Recommendation:** 5

**Clarity, Quality, Novelty And Reproducibility:**

- The paper is relatively well-written. Additional clarifications on some general aspects and assumptions would have been appreciated.
- I'd like to authors to stress further how this proposal departs from other generative models that exploit some prior distributions.


Minor Errors
- [page 1] The de-facto mechanism --> The de-facto standard mechanism
- [page 1] a more border -> a broader
- [page 1] to validate the theoretical results -> why do you need to validate your theory? Use another word.
- [page 4] the the true -> the true


**Details Of Ethics Concerns:**

See the first point in my *Strength And Weaknesses* section.

**Strength And Weaknesses:**

**Presentation**
The paper is generally well-written; There are a few typos and errors but are not affecting the overall readability.
In terms of claims, I would refrain, in the Introduction, to say with such certainty that Membership inference attacks are breaking synthetic data generation models. This is not only an overstretch but is also dangerous as readers may mistrust synthetic data generators, which, in my opinion, provide one of the best alternatives for privately releasing datasets.

**Related work**
The similarity of the proposed method with PrivBayes [1] should be discussed. E.g., where and why, in the proposed approach, the causal representation is different from what done by PrivBayes?

**Technical Questions**
- Can you clarify when the sensitivity of (1) will be smaller than (3) w.r.t. the properties of the causal relationships? Does this hold only when there exists at least one variable with no dependency on y? (see proof of Lemma 1).
- On page 4, under paragraph "why does causality provide any privacy benefit?" the paper claims that "the model will be more stable". Can the authors elaborate further? What assumptions are taken in terms of causal graph dependencies? See also the question I ask above.
- Why are the values of the MI attacks so high?
- The results also do not seem to show that MI attacks can be better overcome (thus better privacy protection) in the causal setting considered. Can you elaborate on your claims?

[1] PrivBayes: Private Data Release via Bayesian Networks. ACM Transactions on Database Systems 42(4):1-41, 2017

**Summary Of The Paper:**

The paper studies the synthesis of privacy-preserving datasets from an interesting angle: It claims knowledge about the causal relationship of the data attributes can simultaneously amplify privacy and improve accuracy.
To shed light on this surprising result, the paper analyzes in which condition a generative model equipped with a causal graph decrease the sensitivity of the data release task, thus increasing accuracy. Surprisingly this also reflects in better model performance against membership inference attacks.
The paper then proposes a VAE-based mechanism to generate causally consistent datasets and provide results over several benchmarks.

**Summary Of The Review:**

The paper studies a very important problem: the release of privacy-preserving data through synthetic generators.
The claims that privacy and accuracy can be both amplified by using additional side information are counter-intuitive to me and require further explanation.

---

> ### Author Response · Authors · 2022-11-15
> **Thank you for your feedback! The comments were very astute!**
>
> Re: comments on presentation – thank you for your feedback. We will present a more nuanced discussion around the vulnerability of synthetic data generation models based on the discussion in https://arxiv.org/abs/2011.07018. Specifically, we will state that several implementation issues result in certain synthetic generation models to be more susceptible to membership inference than others.
>
> Re: comments on related work – to the best of our knowledge, PrivBayes is the only approach that learns a model similar to the causal graph that we aim at inferring. However, the primary shortcoming of the approach is its inability to scale with large datasets. Relying on causal discovery algorithms coupled with generative models is able to overcome this scalability related issues. Additionally, our theoretical results are aimed at showing how faithful causal graphs combined with DP learning provide a privacy amplification, an issue not considered in prior work (utilising distributional knowledge).
>
> Response to technical question 1: No; more details are presented in the “proof of the general case” associated with Lemma 1. The basic idea is that there is a discrepancy b/w causal features X_c and associational features X_a, which can be exploited to ensure the gap in sensitivities.
>
> Response to technical question 2: The results of Peters et al. state that in a scenario with infinite data, a “causal” model will learn a truly faithful representation of the data generating process. If any new data sample (atop the infinite data already seen) is provided to the model (for learning), its parameters will not change. This “no change” property is what we define as stability (i.e., robustness to change in inputs to the learning algorithm).
>
> Response to technical question 3: Re the MI attack efficacy being high – we speculate this to be the case because the attributes are binary.
>
> Response to technical question 4: Our results show that in the causal setting, privacy adversaries are able to exploit some additional information that enables them to launch more effective attacks. This phenomenon is studied in more detail in the recent published work by Baluta et al. (https://arxiv.org/abs/2209.08615)
>
> We hope you find these answers satisfactory. If so, please do consider increasing the score for the submission. If not, kindly let us know what better we can clarify.

---

### Decision · Program_Chairs · 2023-01-20

**Decision:**

Reject

**Justification For Why Not Higher Score:**

The paper does not fall into the ballpark of a borderline paper, thus triaged.

**Justification For Why Not Lower Score:**

N/A

**Metareview: Summary, Strengths And Weaknesses:**

The paper claims new empirical and theoretical results on the stability of causality constraints.  Reviewers generally liked the research direction but find some of the claims unclear. Specifically, the theoretical part and experimental part of the paper seem to be only connected on the very high level.  The theoretical part focuses on analyzing the stability of the fitted parameters with or without a causal graph, when the dataset is generated according to a causal graph.  The connection to the experiments, in which the authors used the standard DP-SGD for training a VAE, is unclear.   It is also unclear how the stability of the fitted parameters implies a "privacy amplification by causality". No claims about differential privacy were given, nor how the parameters of DP improve with causality.

Overall, there is not enough support among reviewers to have a more in-depth discussion to investigate the possibility of acceptance.